# BigO(Bench) - Can LLMs Generate Code with Controlled Time and Space Complexity?

## Abstract

We introduce BigO(Bench), a novel coding benchmark designed to evaluate the capabilities of generative language models in understanding and generating code with specified time and space complexities. This benchmark addresses the gap in current evaluations that often overlook the ability of models to comprehend and produce code constrained by computational complexity. BigO(Bench) includes tooling to infer the algorithmic complexity of any Python function from profiling measurements, including human- or LLM-generated solutions. BigO(Bench) also includes of set of 3,105 coding problems and 1,190,250 solutions from Code Contests annotated with inferred (synthetic) time and space complexity labels from the complexity framework, as well as corresponding runtime and memory footprint values for a large set of input sizes. We present results from evaluating multiple state-of-the-art language models on this benchmark, highlighting their strengths and weaknesses in handling complexity requirements. In particular, token-space reasoning models are unrivaled in code generation but not in complexity understanding, hinting that they may not generalize well to tasks for which no reward was given at training time.

## 1 Introduction

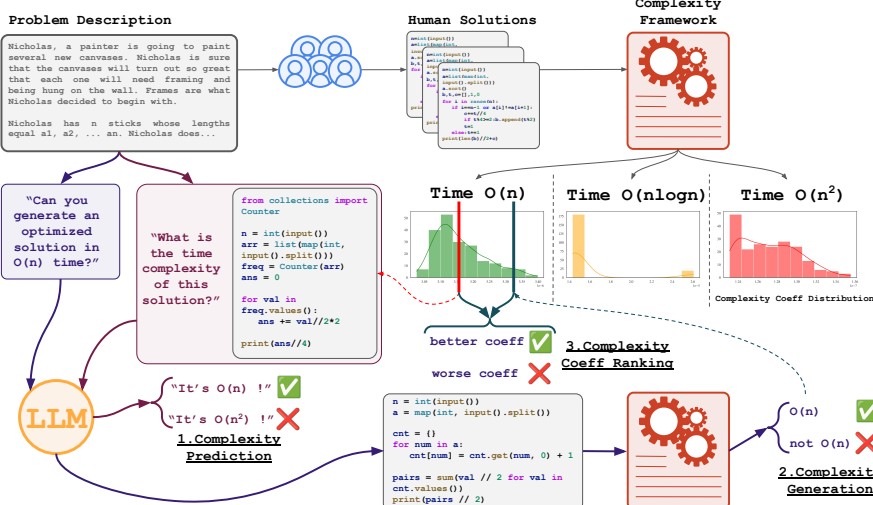

Figure 1: BigO(Bench) framework overview: Given a coding problem and human solutions, the framework evaluates LLMs on three key tasks: (1) predicting time-space complexities of existing solutions, (2) generating new code that meets specified complexity requirements, and (3) ranking solutions against human-written code with similar complexity profiles. The complexity framework automatically validates model outputs by computing runtime distributions and curve coefficients.

A junior developer writes an elegant solution to a coding challenge that passes all test cases, yet fails catastrophically in production. The issue isn't a bug – it's an $O(n^2)$ algorithm processing millions

of records, when an $O(n \cdot \log(n))$ solution could have handled the load effortlessly. As large language models (LLMs) increasingly assist in code generation, their ability to understand and control computational complexity becomes critical. While modern LLMs can generate syntactically correct and functional code with impressive accuracy, our new benchmark BIGO(BENCH) shows that they often struggle with the higher-level reasoning required to optimize time and space complexity – a skill that separates novice programmers from experienced engineers.

Our comprehensive evaluation of state-of-the-art code generation models reveals a concerning gap: while reasoning models like DeepSeek R1 (DeepSeek-AI et al., 2025) achieve above 70% accuracy (pass@1) on programming contests (CODEFORCES[1]), they show significantly weaker performance (4.8%) when tasked with generating solutions under specific complexity constraints (a detailed failure example is in Appendix K). They fare barely better than non-reasoning models at analyzing a function for its complexity, e.g. 6.8% better in complexity prediction vs. Llama 4 Maverick (Meta, 2025). This limitation becomes particularly acute in real-world applications, where scalability and controllable, understandable, performance constraints are often as critical as functional correctness.

Our main contributions are threefold:

- Firstly, to address the challenge mentioned above, we introduce BIGO(BENCH), a novel benchmark for code generation that evaluates a model's understanding of time and space complexities, including runtime and memory profiling measurements for a set of 3,105 coding problems and 1,190,250 solutions from CODE CONTESTS (Li et al., 2022). As shown in Fig. 1, for a given coding challenge and human solution, the model can be queried to a. predict time-space complexities, b. generate code that solves the challenge while adhering to a specified feasible complexity, and c. on top of it ranks better than human solutions of the same challenge and complexity.

- Secondly, we release the code for our complexity inference framework, that takes a Python function and returns time and space complexities. It's a rule-based algorithm based on fuzzing, profiling, and regressing of major complexity classes (including multi-dimensional). This is what we used to produce ground truth labels for BIGO(BENCH), which are statistically significant ground truth performance profiles and not theoretical complexities. This complexity evaluation framework achieves 84% and 82% match (with human annotated theoretical complexity) respectively on time/space complexity test sets.

- Thirdly, we evaluate 14 popular models on our benchmark along fine-tuned ones and compare performance in detail: using our $All@1$ metric, DEEPSEEK-R1 LLAMA70B (DeepSeek-AI et al., 2025) scores best on time complexity prediction (41.4%), while QWEN3 32B(Team, 2025) leads on time complexity generation (6.5%) and LLAMA 3.1 NEMOTRON-ULTRA(Bercovich et al., 2025) on space complexity generation (5.6%).

## 2  RELATED WORK

**Benchmarks for Code Generation**  As the coding skills of LLMs were still limited, benchmarks for code generation originally focused on simple functions and coding challenges, as illustrated by HumanEval (Chen et al., 2021a) and MBPP (Austin et al., 2021) , probably the most famous coding benchmarks. Today, these benchmarks are considered saturated, as top reported $pass@1$ scores lean towards 90-95% success rate. A first area of work has focused on extending, improving quality and correcting these benchmarks, be it with HumanEval+ (Liu et al. (2023) added more tests to pass) or HumanEvalPack (Muennighoff et al. (2024) added more tasks and programming languages).

A different area of research pursues scale and reasoning around code as a way to formulate benchmarks that can challenge LLMs over a longer term. SWE-Bench (Jimenez et al., 2024) and AssistantBench (Yoran et al., 2024) leverage Github as a source of large software development tasks, that do not resemble nor include obvious patterns of reproduction, therefore not solvable with simple fine-tuning on similar data. Nevertheless, the evaluation cost in time and compute is a non-negligible limitation that restrains certain teams from using these benchmarks.

BIGO(BENCH) is a tentative benchmark to integrate challenging notions of reasoning around code into a simple formulation, providing a practical evaluation metric for code LLMs that can easily be used to discriminate and iterate improvements of their coding capabilities.

---

[1]https://codeforces.com/

**Time-Space Complexity Task** Coding interviews for software engineers are centered around small coding challenges to be solved and explained, typically by discussing the time-space complexity of the proposed solutions. Only a few previous works attempted to frame the task of time-space complexity explanation for LLMs. Nevertheless, they all fall short of providing sufficient elements to build a solid benchmark: CoRCoD (Sikka et al., 2019) contains 932 Java code pieces labeled for five time complexity classes without using any LLMs; TASTY (Moudgalya et al., 2023) uses 3000 C++/Python problems across five complexity classes, limited to classification only and just benchmarking small BERT models (Devlin et al., 2019); CodeComplex (Baik et al., 2024) contains 10k Python/Java programs annotated for five time complexity classes as classification only; finally RACE (Zheng et al., 2024a) contains limited test cases ($\sim$100) and only measure runtime similarity between proposed and human solutions.

Most of the work around code efficiency focuses on absolute runtime/memory measurement, scoring code measured as faster/lighter (Qiu et al., 2025; Peng et al., 2025; Huang et al., 2024; Liu et al., 2024; Peng et al., 2024; Du et al., 2024). Whereas absolute runtime remains context-dependent and conditioned on hardware and specific test cases being executed, code complexity reveals the intrinsic comprehension the model has of the underlying algorithmic structure of the code, measuring its efficiency at scale and its asymptotic performance.

In order to improve on the previous attempts, BIGO(BENCH) explores not only time but also space complexity, out of an unconstrained set of classes to capture more various solutions and problems. The benchmark not only studies the classification task but also the more challenging open-framed generation task, so to mimic the real-world thought process of designing a solution for a target complexity. Altogether, this turns out to create a challenging task that wide-used LLMs are benchmarked upon, and hopefully it provides a new perspective on the limitations of current models and their reasoning capabilities around code.

# 3 DYNAMIC COMPLEXITY INFERENCE FRAMEWORK

Throughout this study, complexity refers to worst-case complexity, finding how input growth maximally affects runtime and memory. Python is considered as the only language studied, and therefore complexity can account for python-specific optimizations (e.g. CPython (cpy, 2024) or the compiler) that get reflected in the empirical time and space measures. In the quest of finding the worst case scenario of a snippet of code, natural language constraints on the inputs as detailed in the problem description can be ignored, as long as the program runs and does not fail. Any basic operator like number addition or initialization of an empty list are considered as constant time and space.

**Implementation** The time-space complexity framework is a rule-based algorithm that processes Python functions to infer time and space complexities dynamically. It takes a Python function with example inputs and corresponding dataclass (Section 4.2), processes them, then measures runtime and memory during several executions. From a high-level perspective, the framework increases input sizes using various strategies to assess size impact on execution metrics (runtime, memory). For multi-argument functions, arguments can be expanded independently or together to determine overall complexity, considering interdependencies. Prepared code and expanded inputs run in independent *Bubblewrap* sandboxes (bub, 2024) to prevent harmful side effects. While running, *Cprofiler* is used for runtime measures and *tracemalloc* for memory footprint. Using non-negative least squares curve fitting (Lawson & Hanson, 1976) on each set of measures, the coefficients and residuals of each complexity class are computed. The gold complexity class output for a given set of measures is chosen as the minimizer of the residuals, taking into account a simplicity bias (the more simple the complexity class is, the smaller the simplicity bias). This curve fitting is applied on each set of measures, each corresponding to a different subset of arguments being expanded with a different expansion method. Using ensemble methods, the global complexity of the Python function is computed by aggregating the individual complexity outputs along the different set of measures. Finally, the complexity framework also returns the coefficients of the curve of each elected complexity. These coefficients can be leveraged to rank and classify different optimized Python solutions within the same complexity class. More details are shared on our Github and in Section G.

**Parametrization** The framework involves three main parametrized steps. The first step, *Process Allocation*, handles the multiple ($Code, Expanded\ inputs$) pairs to be run and measured for time

and space, trying to maximize the execution throughput while minimizing its variability and instability. Second, *Execution Measures* consists in leveraging various measuring tools for various ranges of input size values with more or less granularity. Third, *Complexity Fitting* relies on the measures obtained to apply various curve fitting methods, aggregation methods and ensemble methods to form the global complexity formula across all inputs.

Parameters from each of these three groups were optimized towards three metrics of interest: pure accuracy of detecting the correct complexity class, coverage by handling as many code snippets and problems as possible, and self-consistency of outputting stable results over multiple runs and across different compute instances.

## 4 BENCHMARK DATA RELEASE

### 4.1 COMPOSITION

CODEFORCES is an online competitive coding platform that gathers challenging problems to be solved in various programming languages. Humans can submit candidate solutions that are rewarded more the faster and more memory-efficient they are. Using coding problems and solutions from CODEFORCES mostly (and in minority from a few other coding platforms), CODE CONTESTS is a dataset that provides the problem descriptions along with correct and incorrect human solutions.

Figure 2: Distribution of time-space complexity classes across BIGO(BENCH) dataset of 3,105 coding problems. Each problem is included when at least one solution exists with that specific time-space complexity pair. The chart orders classes by computational efficiency, with less common classes grouped under "other".

We annotated data from CODE CONTESTS for time and space complexity to create BIGO(BENCH) using the complexity framework described in Section 3. CODE CONTESTS data was limited to correct solutions (according to public and private tests) written in Python code only, which sum up to 8,139 problems and 1,485,888 solutions in total. In addition, problems that have no working dataclass (see Section 4.2), too few solutions (fewer than 50) or unusual data types are also filtered out. This leads to our general dataset, annotated and released as part of BIGO(BENCH), consisting of 3,105 coding problems and 1,190,250 solutions. Problems are characterized by their difficulty level (A: 942; B: 682; C: 427; D+: 321; Unknown: 733), their algorithmic notions (37 different notions in total) and their inputs (1 to 11 distinct arguments across 32 different data types).

Each solution is annotated by the complexity framework and associated with time and space complexity classes, the corresponding coefficients of the complexity curves and the runtime/memory-footprint measures that were used to infer these attributes. Therefore, each problem gets associated with one or several time-space complexity classes consisting of solutions that have various complexity coefficients, corresponding to different optimization tricks within the same class of complexity. Dataclasses generated for each problem are also released (see Section 4.2).

Time/space complexity test sets are selected among this global pool of problems/solutions by executing a range of post-processing and filtering steps, to provide meaningful metrics by enforcing diversity of classes as well as performance and stability of the framework on the problems. For instance, only problems with several complexity classes are kept; absolute and relative thresholds (to the most popular class of the problem) filter out outliers; complexities with unlikely variable counts or high failure rates are removed; finally solutions with unlikely abstract trees (relative to their complexity) or unstable predictions are withdrawn. The resulting test sets have distinct supports of problems (though there is an overlap of 63 problems), since few problems have diverse classes in both time and space. Moreover they ignore official CODE CONTESTS splits, as the CODE CONTESTS test set lacks sufficient discriminative power, and lacks problems with multiple complexity classes.

The time complexity test set is made out of 311 problems and 640 corresponding solutions covering 11 different classes (the most represented ones being $O(n)$, $O(n \cdot \log(n))$, $O(n^2)$, $O(1)$, $O(n \times m)$ and the least represented $O((n + m)\log(n + m))$). The space complexity test set consists in 308 problems and 636 solutions covering 5 different classes (by order of popularity $O(n)$, $O(1)$, $O(n^2)$, $O(n + m)$, $O(n \times m)$). A training split for fine-tuning purposes is also released. Fig. 2 shares

more details about the distribution of complexity classes in the data being released. It is imbalanced and heavily tailed: linear time complexity represents 38% of all solutions, constant time complexity 20%; for space complexity, distribution is even more skewed with respectively 47% and 25%.

## 4.2 DATACLASS GENERATION

To infer labels with the complexity framework on CODE CONTESTS code snippets, code inputs must be parsable into a dataclass matching Algorithm 1. We define the task of dataclass generation as querying a LLM for such dataclass given the problem description and an example solution. For each generated dataclass `Input`, we introduce two metrics that measure the quality of the dataclass methods `Input.from_str` (converting string stream inputs to argument dictionaries) and `Input.__repr__` (the reverse conversion).

Performance is measured by **CORR**, that accounts for the correction (executability) of the methods `Input.from_str` and `Input.__repr__`, and **BCKTR**, which measures accuracy of the backtranslation (Edunov et al., 2018):

```
input_ == Input.from_str(input_).__repr__()
```

**CORR@k** and **BCKTR@k** are unbiased estimators of performance of **CORR** and **BCKTR** among k samples, following the definition of Chen et al. (2021a). Mathematical definitions of these metrics are provided in Section A.1. Table 1 sums up the benchmark results of the dataclass generation task. LLAMA 3.1 405B INSTRUCT (Dubey et al., 2024) reaches best performance, capable of 58.1% correct backtranslation for one dataclass out of ten attempts per problem. To further boost performance, several passes corresponding to different solutions per problem are performed, thus generating a correct dataclass for 82% of CODE CONTESTS problems.

## 4.3 COMPLEXITY FRAMEWORK PERFORMANCE

**Accuracy**  A human review measured the accuracy of the labels as output by the complexity framework (after post-processing and filtering) compared to the labels assigned by a human. The framework achieves 84% and 82% accuracy on time and space complexity test sets respectively (125 sample split each).

**Coverage**  Fig. 3 measures the ratio of solutions per problem for which the framework fails to predict a label. Whatever the type of complexity, approximately 84% of problems have a fail rate below 30%, and only 4.5% of problems have a fail rate above 0.9, for reasons ranging from incorrect generated dataclass to an edge case not covered by the range of tests the framework performs. Most input types are correctly covered by the framework. Exceptions (e.g. tuple and triple-nested lists) are infrequent.

**Self-consistency**  Relying on empirical runtime and memory measures exposes the framework to stochastic noise affecting prediction reliability. Running the framework 20 times on 10 solutions of every problem and complexity class of the candidate test set, before any filtering based precisely on stability, 91.9% (resp. 89.1%) self-consistency is achieved for time (resp. space) complexity, for a total of 10,130 (resp. 10,520) different code solutions.

Table 1: Comparison of models for generating problem-specific dataclasses that can parse the incoming input streams into each problem's variables, on CODE CONTESTS. All models but CodeLlama 70B Instruct (16k only) (Rozière et al., 2024) use a context window of 32k tokens.

| MODEL | CORR@10 | BCKTR@10 |
|---|---|---|
| CODESTRAL 22B | 63.6 | 54.0 |
| CODELLAMA 34B INSTRUCT | 22.1 | 17.8 |
| CODELLAMA 70B INSTRUCT | 10.3 | 7.9 |
| LLAMA 3.1 8B INSTRUCT | 31.9 | 21.4 |
| LLAMA 3.1 405B INSTRUCT | **70.2** | **58.1** |

---

**Algorithm 1** Dataclass Template

```
@dataclass
class Input:
    var_1: type_1
    var_2: type_2
    ...
    @classmethod
    def from_str(cls, input_: str):
        ...
        return cls(var_1, var_2, ...)
    def __repr__(self):
        ...
        return input_
```

---

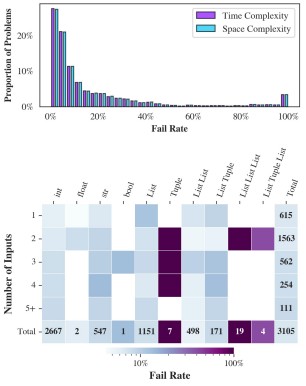

Figure 3: Failure rate analysis of the complexity inference framework. The top plot shows the overall distribution of framework failures across all problems. The bottom heatmap breaks down failure rates by input type and number of distinct inputs.

Table 2: BIGO(BENCH) benchmark results for popular LLMs. **Program Synthesis** checks the correctness of model-generated solutions to given programming problems, not taking into account any complexity requirement. **Complexity Prediction** measures whether a model can find the time-space complexity of an existing code snippet. **Complexity Generation** evaluates whether a model can output a working code snippet to a given problem that meets a time-space complexity requirement. $Pass@k$ considers each complexity class of all problems independently and calculates a macro-average between them. $Best@k$ is a refinement of $Pass@k$, focusing only on the most optimized complexity class for each problem. $All@k$ checks if all complexity classes for each problem are correctly predicted or generated simultaneously, then macro-averages across all problems.

| MODEL | PROG. SYNTHESIS | | COMPLEXITY PRED. | | | COMPLEXITY GEN. | | | |
| --- | --- | --- | --- | --- | --- | --- | --- | --- | --- |
| | PASS @1 | PASS @10 | PASS @1 | BEST @1 | ALL @1 | PASS @1 | PASS @10 | BEST @1 | ALL @1 |
| **TIME** | | | | | | | | | |
| BASELINES | 30.3 | 55.4 | 39.5 | 68.5 | 0.0 | 12.1 | 29.7 | 19.0 | 0.9 |
| LLAMA 3.3 70B | 43.4 | 66.0 | 58.2 | 72.6 | 33.7 | 17.7 | 40.0 | 25.7 | 3.3 |
| LLAMA 4 SCOUT 17Bx16E | 61.7 | 80.1 | 48.7 | 66.3 | 23.0 | 22.8 | 48.0 | 31.8 | 3.5 |
| LLAMA 4 MAVERICK 17Bx128E | 59.3 | 78.9 | 57.4 | 70.8 | 32.8 | 19.9 | 44.6 | 27.0 | 5.3 |
| GEMMA 3 27B | 37.7 | 45.6 | 60.8 | 69.2 | 37.6 | 15.1 | 20.9 | 17.8 | 1.8 |
| CODESTRAL 22B | 23.7 | 47.5 | 56.0 | 67.8 | 33.5 | 10.6 | 26.6 | 14.9 | 1.3 |
| QWEN2.5-CODER 32B | 30.5 | 50.8 | 58.5 | 68.2 | 34.9 | 12.2 | 26.5 | 15.2 | 3.1 |
| GPT-4O | 51.0 | 78.3 | 57.7 | 69.7 | 33.1 | 20.6 | 44.7 | 30.2 | 4.3 |
| O1-MINI | 62.5 | 76.8 | 58.3 | 65.2 | 35.6 | 19.8 | **65.2** | 27.6 | 4.5 |
| DEEPSEEKCODERV2 236B | 44.1 | 65.5 | 54.9 | 68.9 | 29.6 | 19.5 | 38.0 | 27.6 | 3.3 |
| DEEPSEEKV3 671B | 41.4 | 63.6 | 54.4 | 72.4 | 27.1 | 17.7 | 37.7 | 23.0 | 3.4 |
| DEEPSEEKR1 QWEN 32B | 70.1 | 83.7 | 62.2 | 72.7 | 41.1 | 29.0 | 49.9 | 46.3 | 4.8 |
| DEEPSEEKR1 LLAMA 70B | 70.1 | 83.8 | **64.2** | **75.4** | **41.4** | 29.2 | 51.6 | 46.5 | 4.8 |
| LLAMA 3.1 NEMOTRON-ULTRA 253B | **80.0** | **89.4** | 63.1 | 74.5 | 41.1 | **33.5** | 54.9 | **51.8** | 6.1 |
| QWEN3 32B | 70.0 | 81.3 | 61.3 | 70.5 | 39.0 | 29.1 | 53.8 | 43.5 | **6.5** |
| **SPACE** | | | | | | | | | |
| BASELINES | 30.1 | 52.6 | 45.4 | 50.3 | 0.0 | 12.2 | 32.4 | 17.8 | 1.3 |
| LLAMA 3.3 70B | 42.6 | 62.5 | 41.1 | 55.2 | 10.9 | 15.0 | 37.7 | 21.9 | 1.8 |
| LLAMA 4 SCOUT 17Bx16E | 56.7 | 73.5 | 37.4 | 58.8 | 5.2 | 20.0 | 40.1 | 31.1 | 2.2 |
| LLAMA 4 MAVERICK 17Bx128E | 58.4 | 75.7 | 44.6 | 54.5 | 8.9 | 16.8 | 28.2 | 30.4 | 0.8 |
| GEMMA 3 27B | 40.3 | 49.0 | 44.8 | 62.9 | 13.2 | 16.2 | 24.3 | 22.5 | 1.4 |
| CODESTRAL 22B | 25.7 | 47.6 | 44.3 | 62.5 | 10.6 | 11.0 | 29.4 | 16.7 | 1.3 |
| QWEN2.5-CODER 32B | 31.1 | 49.2 | 45.6 | **63.4** | 12.6 | 10.1 | 23.3 | 15.3 | 1.2 |
| GPT-4O | 51.6 | 74.4 | 43.4 | 61.4 | 11.0 | 18.1 | 39.9 | 28.0 | 1.4 |
| O1-MINI | 58.0 | 72.9 | 42.7 | 45.6 | 8.1 | 16.6 | **61.3** | 25.7 | 2.5 |
| DEEPSEEKCODERV2 236B | 43.1 | 63.8 | 44.1 | 59.6 | 8.2 | 16.7 | 34.5 | 25.6 | 1.0 |
| DEEPSEEKV3 671B | 41.8 | 62.5 | 43.5 | 62.6 | 11.2 | 15.0 | 35.4 | 22.6 | 1.6 |
| DEEPSEEKR1 QWEN 32B | 68.0 | 80.6 | 43.2 | 55.0 | 8.1 | 24.8 | 48.6 | 38.6 | 3.1 |
| DEEPSEEKR1 LLAMA 70B | 68.8 | 81.2 | 44.4 | 56.1 | 10.4 | 25.6 | 50.0 | 38.7 | 3.4 |
| LLAMA 3.1 NEMOTRON-ULTRA 253B | **77.7** | **86.4** | 45.2 | 54.7 | 10.3 | **30.4** | 55.5 | **45.3** | **5.6** |
| QWEN3 32B | 65.9 | 77.7 | **47.7** | 58.3 | **15.1** | 25.5 | 47.8 | 39.6 | 5.1 |

## 5 EVALUATION

We use BIGO(BENCH) to evaluate several LLMs commonly used for coding and reasoning tasks: LLAMA 3.3 70B (Dubey et al., 2024), LLAMA 4 models (Meta, 2025), GEMMA 3 27B (Team et al., 2025), CODESTRAL 22B (MistralAI, 2024), GPT-4O (OpenAI et al., 2024b), O1-MINI (OpenAI et al., 2024a), QWEN 2.5-CODER 32B (Hui et al., 2024), DEEPSEEK-CODER-V2 236B (DeepSeek-AI et al., 2024b), DEEPSEEK-V3 671B (DeepSeek-AI et al., 2024a), DEEPSEEK-R1 QWEN and LLAMA distilled (DeepSeek-AI et al., 2025), LLAMA 3.1 NEMOTRON-ULTRA (Bercovich et al., 2025) and QWEN3 32B (Team, 2025). All models are evaluated using their INSTRUCT variant, when available, in a zero-shot fashion (unless otherwise stated). GPT4-O and O1-MINI do not share any estimate on inference compute. Also, O1-MINI returned many empty answers, potentially due to reasoning collapse: we discarded these answers and used only non-empty answers to compute metrics. As a result, its performance can be regarded as an upper-bound optimistic estimate. DEEPSEEK-R1 distilled models used substantially more compute than LLAMA 4 ($\times 2$ compute nodes, $\times 5$ compute time and $\times 16$ generation tokens).

Pure program synthesis performance is also displayed on the same test splits as the rest of the metrics. It is evaluated for $pass@k$ using all public, private and generated tests. For each metric, best values or any values not significantly lower than the best are displayed in boldface. Metrics are macro-averaged first by complexity classes within each problem and then across problems. More details and metric definitions are provided in Section A. [2]

---

[2]One-tailed paired t-tests on 1000 bootstraps samples of the model results evaluate the significance of the superiority of the best model. Any @$k$ metric uses an unbiased estimator based on 20 samples.

## 5.1 TIME-SPACE COMPLEXITY PREDICTION

The first evaluation task of BIGO(BENCH), **Complexity Prediction**, consists in predicting the time and space complexity given a problem description and a human solution. Our baseline for this task is the naive model that always returns $O(n)$, the most frequent class. $Pass@k$ measures the accuracy of finding the correct complexity, using a parsing script that compares the output of the LLM with the ground-truth complexity inferred by the framework; $Best@k$ measures accuracy only across the most optimized complexity class of each problem; $All@k$ requires correct complexity output across all complexity classes at once per problem: the LLM has to correctly output a working solution that meets the complexity requirement for all classes of complexity of the problem. For each metric, $@k$ is the unbiased estimator among k samples, following the definition of Chen et al. (2021a). Metrics are macro-averaged across complexity classes ($Pass@k$), and then across problems ($Pass@k$, $Best@k$ and $All@k$).

Results are displayed in Table 2. A query example, along with an output example of DEEPSEEK-R1 LLAMA 70B, is provided in Section J. More metric definitions are detailed in Section A.2.

## 5.2 TIME-SPACE COMPLEXITY CODE GENERATION

Table 3: Using the complexity framework, the best measured coefficient of the complexity curve, out of 20 attempts, is used to rank LLM-generated code among human solutions from the same problem and time-space complexity class. Ranking is percentile based, n% ranking score amounts for n% human solutions having worse complexity coefficient. If no LLM solution passes correctness tests, ranking score is set to 0. INTERSEC is the subset where all starred (*) models have at least one successful solution.

| MODEL | COEFFICIENT RANKING | | ALL |
|---|---|---|---|
| | FULL | INTERSEC | @1 |
| **TIME** | | | |
| LLAMA 3.3 70B | 33.8 | 65.0 | 2.8 |
| LLAMA 4 SCOUT 17BX16E | 39.0 | 69.9 | 3.4 |
| LLAMA 4 MAVERICK 17BX128E | 43.2 | 72.1 | 3.6 |
| GEMMA 3 27B | 10.4 | 25.7 | 1.6 |
| CODESTRAL 22B | 21.6 | 50.7 | 1.5 |
| QWEN2.5-CODER 32B | 19.7 | 50.4 | 2.2 |
| GPT-4O* | 36.6 | 70.9 | 4.2 |
| O1-MINI* | 26.3 | 78.8 | 3.1 |
| DEEPSEEKCODERV2 236B | 27.7 | 54.1 | 2.8 |
| DEEPSEEKV3 671B | 28.7 | 58.6 | 3.4 |
| DEEPSEEKR1 QWEN 32B* | 38.6 | 79.0 | 4.2 |
| DEEPSEEKR1 LLAMA 70B* | 38.3 | 79.0 | 4.0 |
| LLAMA 3.1 NEMOTRON-ULTRA 253B* | 41.6 | 75.8 | 5.1 |
| QWEN3 32B* | **44.0** | **79.6** | **6.1** |
| **SPACE** | | | |
| LLAMA 3.3 70B | 32.8 | 73.0 | 1.6 |
| LLAMA 4 SCOUT 17BX16E | 34.5 | 78.8 | 2.1 |
| LLAMA 4 MAVERICK 17BX128E | 21.8 | 60.0 | 0.8 |
| GEMMA 3 27B | 17.5 | 49.3 | 1.4 |
| CODESTRAL 22B | 25.2 | 64.6 | 1.2 |
| QWEN2.5-CODER 32B | 20.5 | 68.7 | 0.6 |
| GPT-4O* | 31.6 | 86.3 | 1.3 |
| O1-MINI* | 21.1 | 82.8 | 1.5 |
| DEEPSEEKCODERV2 236B | 26.8 | 68.5 | 1.2 |
| DEEPSEEKV3 671B | 27.2 | 72.3 | 1.3 |
| DEEPSEEKR1 QWEN 32B* | 40.1 | 88.5 | 3.0 |
| DEEPSEEKR1 LLAMA 70B* | 41.6 | **89.5** | 3.3 |
| LLAMA 3.1 NEMOTRON-ULTRA 253B* | **45.4** | 87.7 | **4.6** |
| QWEN3 32B* | 40.5 | 88.2 | 4.3 |

The second task **Complexity Generation** requires the LLM to generate a correct solution to a given problem description that has to respect a feasible time or space complexity requirement. Our baseline for this task is a LLAMA 3.1 70B model that is queried for the same prompts without the complexity requirement. $Pass@k$ measures the accuracy of finding a correct solution, according to public, private and generated tests, that has the correct complexity, as measured by the complexity framework; $Best@k$ and $All@k$ are similarly defined as their counterparts of Section 5.1. Results are displayed in Table 2. An example with DEEPSEEK-R1 LLAMA 70B is provided in Section K.

## 5.3 TIME-SPACE COMPLEXITY COEFFICIENT PERCENTILE RANKING

The third task, **Complexity Coefficient Percentile Ranking**, measures how a generated solution to a given problem, respecting a complexity requirement, ranks among human solutions of the same complexity class and problem. The ranking is performed based on the coefficient of the complexity curve, as measured by the framework: the lower the coefficient, the more flat the complexity curve and the more optimized the solution. Ranking results are given in percentile of the distribution, where a solution of the $nth$ percentile is more optimized than $n\%$ of human solutions. The querying is similar to Section 5.2 with the addition of the requirement "Try to optimize the runtime of your code as much as you can, while respecting the time complexity requirement". See Table 3.

## 5.4 PREDICTION AND GENERATION FINE-TUNING

Using training sets of 2000 problems and 20k code solutions, LLAMA 3.1 70B is being fine-tuned for the first and second task. Time and space generation training sets sums up to 22M tokens, prediction training sets to 18-19M. Each complexity class of each problem includes 10 human examples, filtered following the same steps as the test set creation (see Section 4.1). Models are fine-tuned for 10 epochs in instruct format. See Table 4.

## 6 QUANTITATIVE ANALYSIS

**Understanding time-space complexity** Per Table 2, all LLMs show performance drops on the combined task **Complexity Generation** versus individual tasks **Program Synthesis** and **Complexity Prediction**. Across all tasks, the top performing models remain the reasoning models DEEPSEEK-R1 LLAMA 70B, QWEN3 32B and LLAMA NEMOTRON-ULTRA, achieving 64.2 and 33.5 $Pass@1$ for time prediction and generation, except space prediction, where performance patterns are less clear with smaller differences, as these models tend to overthink extra space complexity despite explicit prompts. Models tend to be even more misled when asked to *"Optimize the solution while respecting the complexity requirement"*, causing 12% average loss in time generation $All@1$ (Table 3), reaching ~30% for GPT-4O and O1-MINI.

At a more granular level, models tend to underperform on non-optimal complexity classes, compared to the most optimized class of every problem, as underlined in Fig. 4. This contradicts human programming patterns, where non-optimized solutions are typically easier than optimal ones, especially for competition problems. In addition, LLMs do not understand that adding dummy pieces of code (e.g., list sorting) could transform a, working linear solution into a less-optimized linearithmic one. In the end, $All@1$ metrics better capture true understanding by uniformly evaluating across optimized and non-optimized solutions, verifying that LLMs do not just stumble upon a solution of the right complexity because they learned by heart the widespread optimized code snippets.

Table 4: BIGO(BENCH) benchmark results for fine-tuned LLAMA 3.1 70B on time-space prediction and generation tasks. Same metrics as in Table 2.

| METHOD | PROG. SYNTH. PASS@1 | PREDICTION | | GENERATION | |
|---|---|---|---|---|---|
| | | TIME ALL@1 | SPACE ALL@1 | TIME ALL@1 | SPACE ALL@1 |
| ZERO-SHOT | **29.6** | 33.8 | 11.9 | **3.1** | **1.8** |
| FEW-SHOT | 28.9 | 33.6 | 12.1 | 2.4 | 1.4 |
| **PREDICTION FINE-TUNING** | | | | | |
| TIME | 27.4 | **36.5** | 6.6 | 2.9 | 1.3 |
| SPACE | 26.6 | 9.0 | **14.0** | 2.4 | 1.4 |
| **GENERATION FINE-TUNING** | | | | | |
| TIME | 23.2 | 34.7 | 12.7 | 1.2 | 1.3 |
| SPACE | 23.4 | 34.6 | 13.0 | 1.5 | 1.4 |

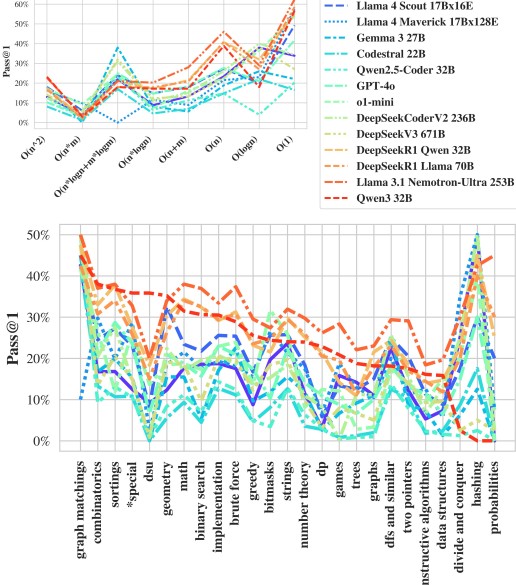

Figure 4: LLM results aggregated by time complexity and algorithmic notions. Scores are $Pass@1$ on Time Complexity Generation.

Top model QWEN3 32B achieves only 6.5 $All@1$ on time generation. These complexity metrics can also be used as a proxy of the understanding of other notions, such as combinatorics, where models explicitly optimized for math reach score higher (Fig. 4). When analyzing these results, one has to keep in mind that the splits were done on the training data of CODE CONTESTS that includes all solutions of the different complexity classes to all problems, already seen by models.

**Token-space reasoning models** Though they largely outperform other LLMs on pure program synthesis, reasoning models such as O1-MINI, DEEPSEEK-R1 and LLAMA NEMOTRON-ULTRA are much closer in terms of performance on complexity-related tasks, and these are even outperformed by QWEN2.5-CODER on space complexity prediction specifically: while the latter obtains 12.6 $All@1$, the former respectively only reach 8.1, 10.4 and 10.3. For this specific case, reasoning models seem to exhibit patterns of overthinking, misunderstanding the notion of extra space complexity, even if clearly described in the prompt. When explicitly prompted in order to understand their failure modes, it turns out that such models are able to recognize all classes of complexity of a problem from a prediction point of view, but when asked to generate them, they fail to return the less optimized classes of complexity, favoring the optimal solution. Not to mention that these models could just easily 'cheat' by tweaking the optimal solution with a dummy sort or nested for-loop, so to transform it, as any programmer could figure it out, as a less-than-optimal solution.

In general, these models struggle with the ambiguity of higher-level reasoning tasks, especially when there is no explicit verifier that they may have been confronted with during their reinforcement, such

as programming tests to pass. This triggers the question of whether they really understand how to 'think' about notions they 'know', or if they only learn by heart patterns of 'thoughts' exhibited by human annotation efforts and training rewards. As all these complexity solutions were, in fact, in their training data, it demonstrates that these highly efficient search models can still fail to recover the correct data points when they were not reinforced for the specific search criteria.

**Developing challenging reasoning benchmarks**   As newly released benchmarks usually quickly saturate, BIGO(BENCH) aims at evaluating high-level reasoning skills that stay out-of-scope of current LLMs, bringing their performance down as displayed by Fig. 5. Table 4 measures that the benchmark remains robust to fine-tuning. In particular, complexity prediction fine-tuning barely improves performance on the same task, and complexity generation fine-tuning even slightly hurt the performance of LLMs. This suggests that learning such high-level reasoning tasks is not effectively captured by standard fine-tuning to learn logical deduction and more convoluted patterns of thoughts. With reasoning models topping benchmarks, new challenging benchmarks are perhaps more about finding out-of-distribution patterns of thinking rather than new data points of existing reasoning scenarios. It was especially difficult to design BIGO(BENCH) without any human labels available, as the qualification level required from annotators was not reachable. But this is also probably why the models had been underexposed to this reasoning task, therefore making it challenging for them.

**Limitations**   The complexity framework itself is prone to errors, as for specific problems it can potentially fall upon worst-complexity edge cases. In addition, the measures on which the complexity prediction is based remain noisy, still relying on real CPU runtimes and using statistical measuring tools; they could potentially become more reliable with the help of virtual CPU cores.

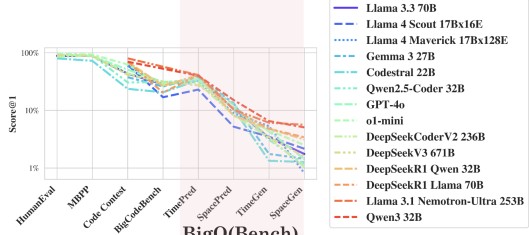

Figure 5: Model scores per coding benchmark: HUMANEVAL, MBPP and BIGCODEBENCH main metrics are all $Pass@1$; for BIGO(BENCH), we display $All@1$ results.

Although we fine-tune LLAMA models, we did not use advanced multiturn prompting (Zheng et al., 2024b) nor further reinforcement (Gehring et al., 2025). Proximal Policy Optimization (Schulman et al., 2017) could help refine LLMs for these tasks. Human annotations could also help models reason better for these tasks. Finally, the coding problems and the framework remain limited to Python. Mixing other languages such as C++ and Java could measure cross-languages optimization strategies.

# 7   CONCLUSION

In this work, we introduced BIGO(BENCH), a novel benchmark of LLMs on code generation, focusing on their understanding of time and space complexities when producing code. It consists in three tasks: first, given a coding challenge, predicting the time-space complexity of a given corresponding solution; second, for a given challenge and time or space complexity, generating a solution that solves the challenge while fulfilling the complexity requirements; third, optimizing the coefficient of the complexity cure compared to the human distribution. This benchmark is supported by the release of time-space complexity labels corresponding to 3,105 coding problems and 1,190,250 corresponding solutions from CODE CONTESTS. In addition, we developed and release the code of a complexity framework, capable of dynamically inferring the time-space complexity of a given snippet of code, used to automatically evaluate any synthetic snippet of code and therefore the generation of LLMs conditioned on a given complexity trade-off. Finally, we benchmark 14 LLMs considered as the top coding assistants and analyse their performance.

BIGO(BENCH) is a challenging benchmark with current top scores belonging to the reasoning models DEEPSEEK-R1 LLAMA 70B, QWEN3 32B and LLAMA NEMOTRON-ULTRA, achieving up to 6.5% and 5.6% $All@1$ on the time-space complexity generation tasks. Even when fine-tuning a LLAMA 3.1 70B model, performance increases only marginally, only on prediction tasks. We hope this benchmark can keep challenging upcoming models and help guide the development of new models towards better understanding of coding abstract notions beyond pure code generation.

## 8 ETHICS STATEMENT

BIGO(BENCH) does not include any model release, and therefore there is no risk for model misuse. The data used for BIGO(BENCH) comes from an already existing public dataset (Li et al., 2022). Any code execution is to be done within sandboxes, as documented and provided in our associated Github repository, using the *Bubblewrap* library (bub, 2024), to avoid any harmful side effects of the code being run.

Furthermore, we establish guidelines for users to follow when utilizing the benchmark, including the requirement to report any potential misuse or harmful applications, and to provide transparency in their use of the benchmark data and results. More details are provided in the Code of Conduct that we publish alongside our Github repository. We also establish a mechanism for users to report any concerns or issues related to the benchmark, and to provide feedback on how to improve the benchmark's safety and responsibility, on top of any potential security issues: this is detailed further in the security policy attached to our code repository as well.

## 9 REPRODUCIBILITY STATEMENT

To ensure reproducibility of the work presented in this paper, we provide comprehensive details on the different steps of our study. The complexity framework implementation is detailed in Section 3 with technical specifics in Section G. Our dataset construction and pre-processing procedures are documented in Section 4. Benchmark design methodologies and theoretical foundations for our evaluation metrics are described in Section A, while time-space complexity definitions are formalized in Section B. Ablation studies validating our benchmark design choices are presented in Section C.

The computational infrastructure and experimental setup used for our evaluations are documented in Section H. Main experimental results are presented in Section 5, with extended multi-sample analysis in Section D. Representative examples of benchmark prompts for both complexity prediction and generation tasks are provided in Sections J and K. All open-source models and their repositories are listed in Section I, while we also evaluate several closed-source models accessible through vendor platforms. Upon acceptance, we will release our complete codebase, dataset, and interactive leaderboard to facilitate future research.

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

# A    FORMAL METRIC DEFINITIONS

## A.1    DATACLASS GENERATION METRICS

In order to measure the quality of the dataclasses as generated by a LLM, we introduce two metrics **CORR** and **BCKTR**, used over a test set of code problems and solutions to compare performance of LLMs on the dataclass generation task.

We introduce the following variables, and the structure of a problem dataclass:

- $P$ = Problem description
- $S$ = Example solution
- $D$ = LLM-generated dataclass (from $P$ and $S$), that has the following class template (see Algorithm 1):
$$D = \begin{cases} D.\text{from\_str}(\cdot) : \text{str} \to \text{dict} \\ D.\text{\_\_repr\_\_}() : \text{dict} \to \text{str} \end{cases}$$
- input = Input string, that corresponds to the input of an Input/Output test case pair of the code problem
- $D.\text{from\_str}(\text{input})$ = Dictionary of parsed arguments
- $D.\text{\_\_repr\_\_}()$ = String representation of parsed arguments
- $\mathcal{P}$: Distribution over problems $P$
- $\mathcal{S}(P)$: Distribution over human solutions $S$ for problem $P$
- $\mathcal{D}_{LLM}(P, S)$: LLM's output distribution for dataclass $D$ given $P$ and $S$.
- $\mathcal{I}(P)$: Set of test case inputs for problem $P$.

Based on these definitions, we define the following two metrics:

1. **Correction (Corr)**: Executability of both methods
$$\text{Corr} = \mathbb{E}_{P \sim \mathcal{P}, \, S \sim \mathcal{S}(P), \, D \sim \mathcal{D}_{LLM}(P,S)} \mathbb{I}(D.\text{from\_str}(\cdot) \text{ and } D.\text{\_\_repr}() \text{ are executable})$$

2. **Backtranslation Accuracy (BckTr)**: Round-trip consistency

$$\text{BckTr} = \mathbb{E}_{P \sim \mathcal{P}, \, S \sim \mathcal{S}(P), \, D \sim \mathcal{D}_{LLM}(P,S)} \mathbb{I} \left( \bigcap_{\text{input} \in \mathcal{I}(P)} (\text{input} == D.\text{from\_str}(\text{input}).\text{\_\_repr}()) \right)$$

In practice, we will want an approximation of these metrics, especially as we are operating under a fixed sampling budged for the LLM. We define a sampling budget parametrized by $n$, the number of times we will be sampling a dataclass $D$ from $\mathcal{D}_{LLM}(P, S)$. Each problem is sampled with equal probability, and for the sake of limiting the compute budget only the first human solution of the attached set of solutions $\mathcal{S}(P)$ will be used to compute the metrics. Then, for each $(P, S)$, we use the model to generate $n$ samples $D_1, \dots, D_n \sim \mathcal{D}_{LLM}(P, S)$.

We generalize the above metrics to the case of k-success, where we want to record the probablity that at least one of $k$ generated dataclasses $\{D_i\}_{i=1}^{k}$ satisfies the metric, leveraging our compute budget of $n$ possible samples (in practice we take $n = 2 \times k$) to have the best unbiased estimator (we use Codex (Chen et al., 2021b) $pass@k$ estimator).

This leads us to the following definitions of the estimators **Corr**$@k$ and **BckTr**$@k$:

1. **Corr**$@k$ estimator that at least one of k LLM attempts satisfies dataclass correction
$$\text{Corr} = \mathbb{E}_{\substack{P \sim \mathcal{P} \\ S_1 \in \mathcal{S}(P)}} \left[ 1 - \frac{\binom{n-c}{k}}{\binom{n}{k}} \right], \quad \text{where} \begin{cases} c = \sum_{i=1}^{n} \mathbb{I}(\text{Corr is satisfied for } D_i), \\ D_i \sim \mathcal{D}_{LLM}(P, S_1). \end{cases}$$

2. **BckTr**$@k$ estimator that at least one of k LLM attempts satisfies backtranslation of the dataclass methods
$$\text{Corr} = \mathbb{E}_{\substack{P \sim \mathcal{P} \\ S_1 \in \mathcal{S}(P)}} \left[ 1 - \frac{\binom{n-c}{k}}{\binom{n}{k}} \right], \quad \text{where} \begin{cases} c = \sum_{i=1}^{n} \mathbb{I}(\text{BckTr is satisfied for } D_i), \\ D_i \sim \mathcal{D}_{LLM}(P, S_1). \end{cases}$$

## A.2 COMPLEXITY METRICS

### A.2.1 COMPLEXITY PREDICTION METRICS

The first evaluation task, **Complexity Prediction**, involves predicting the time and space complexity of a problem given its description and a human solution. The metrics for this task are defined as follows:

- **Pass@k**: Measures the accuracy of correctly predicting the complexity class for each problem, macro-averaged across complexity classes and then across problems. The unbiased estimator among $k$ samples is used.

- **Best@k**: Measures accuracy only for the most optimized complexity class of each problem, using the unbiased estimator among $k$ samples.

- **All@k**: Requires correct complexity output across all complexity classes simultaneously for each problem, using the unbiased estimator among $k$ samples.

We share below more formal definitions of these metrics. First, we define the following setup for the task, on top of the general set up of coding problems involving problems $P \sim \mathcal{P}$ and corresponding human solutions $S \sim \mathcal{S}(P)$ as introduced in Section A.1:

- $\mathcal{C}$: Set of complexity classes (e.g., time, space).

- For each problem $P$, $\mathcal{C}(P) \subseteq \mathcal{C}$: Subset of complexity classes relevant to $P$. This set is determined in our case using the complexity framework on the ground truth human solutions, and post-processing the distribution of complexity classes (typically to remove any obvious outlier).

- $\mathcal{H}(P, c)$: Set of human solutions for problem $P$ and complexity class $c \in \mathcal{C}(P)$.

- $S'$: LLM-generated solution containing the predicted complexity for the given human code $S$ and problem $P$. This solution is generated by the LLM $S' \sim \mathcal{D}_{LLM}(P, S)$.

- $G$ the function that extracts the ground-truth label assigned to the human code, the very same complexity that we are trying to predict using the LLM.

- parse$(S')$: Parsing function that extracts the predicted complexity of class $c'$ from $S'$.

- $\mathbb{I}_{\text{correct}}(S', S, P)$: Indicator function equal to 1 if parse$(S') = G(S)$, else 0.

That said, we define the above introduced metrics as the following:

1. **Pass@k**: Measures correctness for each complexity class of each problem independently, averaged over all classes and problems. Correctness of having at least one correct solution out of $k$ attempts, using a sampling budget of $n > k$ (usually using $n = 2 \times k$):

$$\text{Pass@k} = \mathbb{E}_{P \sim \mathcal{P}} \left[ \frac{1}{|\mathcal{C}(P)|} \sum_{c \in \mathcal{C}(P)} \left[ 1 - \frac{\binom{n - c_c}{k}}{\binom{n}{k}} \right] \right],$$

where $c_c$ is the count of times the LLM found the correct solution, that is to say labeled the complexity class of the human solution correctly:

$$c_c = \sum_{i=1}^{n} \mathbb{I}_{\text{correct}}(S'_i, S_1, P),$$

and $S_1$ is a human solution from the same problem and complexity class $S_1 \in \mathcal{H}(P, c)$.

2. **Best@k**: Evaluates correctness only for the best (e.g., most efficient) complexity class per problem, in particular the probability of getting one correct answer among k answers:

$$\text{Best@k} = \mathbb{E}_{P \sim \mathcal{P}} \left[ 1 - \frac{\binom{n - c_{c_{\text{best}}}}{k}}{\binom{n}{k}} \right],$$

where $c_{c_{best}}$ is the count of times the LLM found the correct solution, that is to say labeled the complexity class of the human solution correctly, only for the best class of complexity $c_{best}$ for each problem:

$$c_{\text{best}} = \sum_{i=1}^{n} \mathbb{I}_{\text{correct}}(S'_i, S_1, P),$$

with $c_{\text{best}}(P)$ is the most optimized complexity class for $P$, and $S_1 \in \mathcal{H}(P, c_{\text{best}}(P))$

3. **All@k**: Joint accuracy across all complexity classes. It requires correctness for all complexity classes simultaneously:

$$\text{All@k} = \mathbb{E}_{P \sim \mathcal{P}} \left[ 1 - \frac{\binom{n - c_{\text{all}}}{k}}{\binom{n}{k}} \right],$$

where

$$c_{\text{all}} = \sum_{i=1}^{n} \mathbb{I} \left( \bigcap_{c \in \mathcal{C}(P)} \mathbb{I}_{\text{correct}}(S'_i, S_c, P) \right)$$

with $S_c \in \mathcal{H}(P, c)$.

These metrics all use the unbiased estimator $1 - \dfrac{\binom{n - c_{\cdot}}{k}}{\binom{n}{k}}$ from Chen et al. (2021b), where $n$ is the total number of samples, and $c_{\cdot}$ is the count of valid samples, usually parametrized by a complexity class for instance.

### A.2.2 COMPLEXITY GENERATION METRICS

The second task, Complexity Generation, involves generating solutions that meet specific complexity requirements. The metrics are defined as follows:

- **Pass@k**: Measures performance of generating a correct solution that meets each complexity class requirement independently, averaged over all classes and problems.

- **Best@k**: Measures generation correctness for the most optimized complexity class per problem.

- **All@k** Requires the solution to meet all complexity classes simultaneously for each problem.

For each problem $P \sim \mathcal{P}$:

- $\mathcal{C}(P)$: Set of complexity classes (e.g., time and/or space) that exist for a particular problem.

- $S'$: Generated solution by the LLM. We can write $S'(P, c)$ or $S' \sim \mathcal{D}_{LLM}(P, c)$ to precise that the generation of the LLM is conditioned on a particular problem $P$ and a requested complexity class $c$ for the solution.

- $\mathbb{I}_{\text{correct}}(S', P)$: 1 if $S'$ passes all tests associated with problem $P$ which means it is evaluated as a correct solution to the coding challenge, 0 otherwise (therefore this does not take into account the complexity of the generated solution).

- $\mathbb{I}_{\text{class}}(S', c, P)$: 1 if $S'$ meets complexity class $c \in \mathcal{C}(P)$, 0 otherwise. This does not depend on the correctness of the solution with respect to the coding challenge $P$ alone. So a solution that compiles, runs and produces wrong results can potentially meet the complexity requirement.

In this context, we can define the following metrics associated with the task of complexity generation:

- **Pass@k**:
$$\text{Pass@k} = \mathbb{E}_{P \sim \mathcal{P}} \left[ \frac{1}{|\mathcal{C}(P)|} \sum_{c \in \mathcal{C}(P)} \left( 1 - \frac{\binom{n - c_c}{k}}{\binom{n}{k}} \right) \right],$$
where $c_c = \sum_{i=1}^{n} \mathbb{I}_{\text{class}}(S'_i, c, P) \times \mathbb{I}_{\text{correct}}(S'_i, P)$.

- **Best@k**:
$$\text{Best@k} = \mathbb{E}_{P \sim \mathcal{P}} \left[ 1 - \frac{\binom{n - c_{\text{best}}}{k}}{\binom{n}{k}} \right],$$
where $c_{\text{best}} = \sum_{i=1}^{n} \mathbb{I}_{\text{class}}(S'_i, c_{\text{best}}(P), P) \times \mathbb{I}_{\text{correct}}(S'_i, P)$, and $c_{\text{best}}(P)$ is the most optimized class in $\mathcal{C}(P)$.

- **All@k**:
$$\text{All@k} = \mathbb{E}_{P \sim \mathcal{P}} \left[ 1 - \frac{\binom{n - c_{\text{all}}}{k}}{\binom{n}{k}} \right],$$
where $c_{\text{all}} = \sum_{i=1}^{n} \prod_{c \in \mathcal{C}(P)} \mathbb{I}_{\text{correct}}(S'_{i,c}, P) \times \mathbb{I}_{\text{class}}(S'_{i,c}, c, P)$ with $S'_{i,c} \sim \mathcal{D}_{LLM}(P, c)$.

### A.2.3 COMPLEXITY RANKING METRICS

Concerning the Complexity Ranking Task, as explained in the main manuscript, the prompt is modified so to query the model for the complexity generation task, with an optimization addition - the model needs to generate correct code, with correct complexity, as optimized as possible while adhering to this complexity requirement.

To formalize the metrics of this task, we add the following elements to the formalism defined in the previous subsections:

- $\mathcal{H}(P, c)$: Set of human solutions for problem $P$ and complexity class $c \in \mathcal{C}(P)$.

- coeff$(S, P, c)$: Complexity coefficient of solution $S$ for problem $P$ and class $c$ (lower is better).

- best_coeff$_{LLM}(P, c)$: Best coefficient from $n = 20$ LLM-generated solutions for $P$ and $c$. This is obtained by sampling $S'_1, S'_2, \ldots S'_n \sim \mathcal{D}_{LLM}(P, c)$ and ranking coeff$(S'_1, P, c)$, coeff$(S'_2, P, c)$, $\ldots$ coeff$(S'_n, P, c)$.

- $\mathbb{I}_{\text{correct}}(S', P)$ has the exact same definition as for the complexity generation task. It measures whether the generated solution is correct for a given problem given its I/O test cases.

- $\mathbb{I}_{\text{class}}(S', c, P)$ has the exact same definition as for the complexity generation task. It measures whether a generated solution respects the complexity class it was conditioned upon.

We now introduce the following metrics. **Pass@k**, **Best@k** and **All@k** have the same definitions as in the complexity generation task:

- **Pass@k**:

$$\text{Pass@k} = \mathbb{E}_{P \sim \mathcal{P}} \left[ \frac{1}{|\mathcal{C}(P)|} \sum_{c \in \mathcal{C}(P)} \left( 1 - \frac{\binom{n - c_c}{k}}{\binom{n}{k}} \right) \right],$$

where $c_c = \sum_{i=1}^{n} \mathbb{I}_{\text{class}}(S'_i, c, P) \times \mathbb{I}_{\text{correct}}(S'_i, P)$.

- **Best@k**:

$$\text{Best@k} = \mathbb{E}_{P \sim \mathcal{P}} \left[ 1 - \frac{\binom{n - c_{\text{best}}}{k}}{\binom{n}{k}} \right],$$

where $c_{\text{best}} = \sum_{i=1}^{n} \mathbb{I}_{\text{class}}(S'_i, c_{\text{best}}(P), P) \times \mathbb{I}_{\text{correct}}(S'_i, P)$, and $c_{\text{best}}(P)$ is the most optimized class in $\mathcal{C}(P)$.

- **All@k**:

$$\text{All@k} = \mathbb{E}_{P \sim \mathcal{P}} \left[ 1 - \frac{\binom{n - c_{\text{all}}}{k}}{\binom{n}{k}} \right],$$

where $c_{\text{all}} = \sum_{i=1}^{n} \prod_{c \in \mathcal{C}(P)} \mathbb{I}_{\text{correct}}(S'_{i,c}, P) \times \mathbb{I}_{\text{class}}(S'_{i,c}, c, P)$ with $S'_{i,c} \sim \mathcal{D}_{LLM}(P, c)$.

- **COEFF**: Percentile ranking against human solutions of the same problem and complexity class. It compares the LLM's best coefficient to human solutions in the same class. This metric uses a fixed sampling budget of size $n$ (in our experiments we take $n = 20$):

$$\text{COEFF} = \mathbb{E}_{P \sim \mathcal{P}, c \in \mathcal{C}(P)} \left[ \text{rank}_{LLM}(P, c) \right],$$

where:

$$\text{rank}_{LLM}(P, c) = \begin{cases} \frac{|\{h \in \mathcal{H}(P,c) | \text{coeff}(h, P, c) > \text{best\_coeff}_{LLM}(P, c)\}|}{|\mathcal{H}(P,c)|} \times 100 & \text{if } \exists S' \text{ valid,} \\ 0 & \text{otherwise.} \end{cases}$$

In addition, we also introduce COEFF$_{intersect}$ (and distinguish it from COEFF by denoting the latter one as COEFF$_{full}$) as the expectation used in the definition of COEFF restricted

to a subset of problems and complexity classes. For the set $\mathcal{M}^*$ of stared models of the complexity ranking table, we define the set of problems $\mathcal{P}'$ as the collection of problems $P$ from $\mathcal{P}$ where there exists at least one complexity class $c$ associated with $P$, to that for every model $M$ in the set $\mathcal{M}^*$, sampling $n$ solutions from $M$ on $P$ and $c$ yields at least one valid solution. Formally:

$$\mathcal{P}' = \left\{ P \in \mathcal{P} \mid \exists c \in \mathcal{C}(P), \forall M \in \mathcal{M}^*, \sum_{i=1}^{n} \mathbb{I}_{\text{correct}}(S_i', P) \times \mathbb{I}_{\text{class}}(S_i', c, P) \geq 1 \right.$$
$$\left. \text{where } S_i' \sim \mathcal{D}_M(P, c) \right\}$$

Similarly, we define the set of corresponding complexity classes $\mathcal{C}'(P)$ where all models are indeed producing at least one correct sample code. For any $P \in \mathcal{P}'$, we define $\mathcal{C}'(P)$:

$$\mathcal{C}'(P) = \left\{ c \in \mathcal{C}(P) \mid \forall M \in \mathcal{M}^*, \sum_{i=1}^{n} \mathbb{I}_{\text{correct}}(S_i', P) \times \mathbb{I}_{\text{class}}(S_i', c, P) \geq 1 \right.$$
$$\left. \text{where } S_i' \sim \mathcal{D}_M(P, c) \right\}$$

Based on that, $\text{COEFF}_{intersec}$ can formalized as:

$$\text{COEFF}_{intersect} = \mathbb{E}_{P \sim \mathcal{P}', \, c \in \mathcal{C}'(P)} \left[ \text{rank}_{LLM}(P, c) \right]$$

where we are using the above definition of $\text{rank}_{LLM}(P, c)$.

Note that the COEFF metric uses the best coefficient from $n$ (20 in practice) LLM attempts and ranks it against human solutions in the same complexity class. A higher percentile means the LLM's solution is more optimized than most human solutions. The **Intersec** subset ensures comparisons are only made when all models have at least one valid solution, avoiding skewed rankings from partial failures. We are selecting models that are good enough in the first place so to make the subset viable and interesting (it has to contain enough samples in the first place). Finally, the unbiased estimator $1 - \dfrac{\binom{n-c}{k}}{\binom{n}{k}}$ remains consistent with prior work (Chen et al., 2021b).

# B  TIME AND SPACE COMPLEXITY DEFINITIONS

## B.1  INTRODUCTION ON THE NOTION OF COMPLEXITY

When writing software, it is crucial to understand how our algorithms perform with different input sizes. Complexity analysis helps us:

1. Predict performance on large inputs, so to understand how the code would behave in runtime and memory footprint on large untested inputs.

2. Compare different algorithms objectively, not on a few example cases but in a generalized case.

3. Identify optimization opportunities.

Big O notation describes how an algorithm's requirements (time or space) grow as input size increases. It focuses on:

- Worst-case scenario: Maximum required resources
- Growth rate: How needs scale with input size $n$
- Upper bound: Simplified representation of complexity

When analyzing time complexity, we simplify our analysis by focusing on the fundamental growth pattern rather than exact measurements. Constant factors are disregarded because they do not affect the overall growth rate - an algorithm that takes $2n$ operations and one that takes $100n$ operations are both considered $\mathcal{O}(n)$ since their linear scaling behavior is identical. Similarly, lower-order terms become insignificant as input sizes grow large; for example, $\mathcal{O}(n^2 + n)$ simplifies to $\mathcal{O}(n^2)$ because the quadratic term dominates the growth pattern. Ultimately, we focus on identifying the dominant term - the component of the complexity expression that grows fastest with input size - as this determines the algorithm's scalability characteristics in the worst-case scenario.

## B.2  ALGORITHMIC EXAMPLES

Time complexity measures how the number of operations grows with input size. Below are listed a few examples of Python codes that belong to various time complexity classes:

**Constant Time (O(1))**
Algorithms with constant time complexity execute in the same time regardless of input size. This is achieved through direct access operations, like retrieving the first element of an array:

```python
def get_first_element(arr):
    return arr[0]  # Execution time remains constant
```

**Logarithmic Time (O(log n))**
Logarithmic algorithms reduce the problem size exponentially with each step. Binary search demonstrates this by halving the search space each iteration:

```python
def binary_search(arr, target):
    low, high = 0, len(arr)-1
    while low <= high:
        mid = (low + high) // 2
        if arr[mid] == target:
            return mid
        low = mid + 1 if arr[mid] < target else high = mid - 1
    return -1
```

**Linear Time (O(n))**
Linear algorithms scale directly with input size, requiring a single pass through all elements:

```
def sum_elements(arr):
    total = 0
    for num in arr:  # Processes each element exactly once
        total += num
    return total
```

**Linearithmic Time (O(n log n))**

This complexity combines linear and logarithmic growth, seen in efficient sorting algorithms like merge sort:

```
def merge_sort(arr):
    if len(arr) > 1:
        mid = len(arr) // 2
        left = merge_sort(arr[:mid])   # Divide phase
        right = merge_sort(arr[mid:])  # O(log n) divisions
        return merge(left, right)      # O(n) merging
    return arr
```

**Quadratic Time (O(n²))**

Nested iterations over input data characterize quadratic complexity, as seen in bubble sort:

```
def bubble_sort(arr):
    n = len(arr)
    for i in range(n):                 # Outer loop (O(n))
        for j in range(n-i-1):         # Inner loop (O(n))
            if arr[j] > arr[j+1]:      # Comparison operation
                arr[j], arr[j+1] = arr[j+1], arr[j]
    return arr
```

**Exponential Time (O($2^n$))**

Algorithms with exponential complexity double their runtime with each new element, exemplified by naive Fibonacci calculation:

```
def fib(n):
    if n <= 1:  # Base case
        return n
    return fib(n-1) + fib(n-2)  # Recursive branching
```

**Factorial Time (O(n!))**

The most resource-intensive complexity class grows factorially, demonstrated by permutation generation:

```
from itertools import permutations
def all_permutations(arr):
    return list(permutations(arr))  # Generates n! combinations
```

Space complexity examines the growth of memory usage with input size. Key considerations include auxiliary space, input storage, and recursion stack usage. Below are a few examples in Python code of various space complexity classes:

**Constant Space (O(1))**

Algorithms using fixed memory regardless of input size:

```
def swap(a, b):
    a, b = b, a  # Temporary variables only
    return a, b
```

**Linear Space (O(n))**

Memory usage scales directly with input size, as in array copying:

```python
def create_copy(arr):
    copy = [x for x in arr]    # New array of same size
    return copy
```

**Quadratic Space (O(n²))**

Memory grows with input squared, typical for 2D arrays:

```python
def create_matrix(n):
    return [[0 for _ in range(n)] for _ in range(n)]   # n x n grid
```

**Logarithmic Space (O(log n))**

Efficient divide-and-conquer algorithms use logarithmic space:

```python
def binary_tree_depth(node):
    if not node:
        return 0
    return 1 + max(   # Single recursive branch at a time
        binary_tree_depth(node.left),
        binary_tree_depth(node.right)
    )
```

**Recursive Space Considerations**

Recursive implementations have hidden stack costs. The Fibonacci example shows O(n) space despite $O(2^n)$ time:

```python
def recursive_fib(n):
    if n <= 1:
        return n
    return recursive_fib(n-1) + recursive_fib(n-2)
```

Each recursive call adds a stack frame, creating O(n) space complexity from the maximum recursion depth.

B.3  BIG-O VERSUS BIG-THETA

It is important to notice that not only does big-O notation exist but also big-Theta and big-Omega. In mathematical analysis, these notions are clearly defined over a single-variable function (or at least a function on an input space with a well-defined ordering) as respectively an upper bound, a tight both upper and lower bound, and a lower bound. The problem is that these notions do not clearly translate to the domain of programming given that in many cases programs take a variety of inputs that are not clearly ordered (this is the case for example for a function that takes an integer and a dictionary, or a function that takes an integer and a list where it is not clear whether a list gets "bigger" when elements of the list grow in size versus when the list itself grows, not to mention how the content of the list input may intertwine with the integer input). For this reason, programming refers to "running cases", where a running case can be seen as a generator of inputs parametrized by a single variable (the size), and can be denoted as "best-running case" when running the program on this case across many sizes leads to the best time execution (or memory footprint) curve. For a well-defined running case $g$ of the code program $f$, this enables to close the gap with the well-defined mathematical analysis definition as $f \circ g$ is now a single-variable function of a correctly ordered input space.

In this context, programming does have to take into account best, average and worst running cases when talking about big-O, big-Theta and big-Omega notations. Talking about general big-Theta behavior of a program is not defined in many programming cases, as this would require the program to have the same big-Theta behavior across all running cases, if we were to generalize the definition. For example, Quick Sort has a best running case complexity in $O(1)$ and worst running case complexity in $O(n^2)$, which means no clear big-Theta behavior is generally defined for this program: in

this case, the query "predict the big-Theta complexity of this program" is not clearly defined. For this reason, we chose to adhere to the commonly accepted definition in programming of assuming that:

- BigO is used to describe the worst running case complexity of a piece of code, that is to say the tightest possible lower bound of this running case.

- BigTheta is used to describe a tight bound, that exists only when the algorithm does behave the same on any type of input.

- BigOmega is used to describe the best running case complexity, that is to say the tightest possible upper bound of this running case.

On top of that, we underline that the goal of the BIGO(BENCH) benchmark is also to reflect the usability of LLMs in practice, and provide an accurate measure of the performance of LLMs when being queried by daily users. Choosing a prompt that does not involve too many definitions, though it may leave some unclarities and ill-defined terms, ensures we better capture how the LLM performance will be perceived by users.

More ablations on the task prompts, including how we refer to the notion of complexity, are presented in Section C.

## C    ABLATIONS ON TASK PROMPTS

We ran ablations on the prompts, for each task, comparing the performance on a reasoning (taking QWEN QWQ 32B) and a non-reasoning model (LLAMA 3.3 70B). The following subsections provide the types of prompts that were used (every time the example corresponds to the time complexity variant, given that the space complexity variant can easily be derived from it; similarly, given the complexity prediction variant, the complexity generation variant can also be derived in a simple manner).

### C.1    ORIGINAL BIGOBENCH PROMPT - TIME COMPLEXITY PREDICTION

Provide the time complexity for the following competitive programming question and corresponding solution.

When analyzing the complexity of an algorithm, consider the worst-case scenario where all possible input combinations are tried, given the following conditions: 1. the inputs must adhere to the specified data types of the problem; 2. the inputs should not cause the code to crash or exit on an exception; 3. the inputs do not necessarily need to satisfy additional constraints that are potentially mentioned in the problem statement; 4. calling input() does not consume runtime nor memory, but of course any operations on top of it or afterwards will be counted towards runtime and memory footprint; 5. Anything printed gets added to the memory. You can take advantage of Python-specific optimizations provided by the underlying CPython interpreter or compiler to achieve the desired complexity, and you must account for them when analyzing the complexity.

Here is the programming question: context.

Here is the corresponding Python solution: code_content.

Please ignore any constraints on the input sizes that may have been previously mentioned in the problem description. Compute the big-O complexity as if inputs can be as large as possible.

Output the big-O time complexity only, no explanation needed, no other words needed.

### C.2    BIG-THETA VARIANT - TIME COMPLEXITY PREDICTION

Provide the time complexity for the following competitive programming question and corresponding solution.

When analyzing the complexity of an algorithm, consider the worst-case scenario where all possible input combinations are tried, given the following conditions: 1. the inputs must adhere to the specified data types of the problem; 2. the inputs should not cause the code to crash or exit on an exception; 3. the inputs do not necessarily need to satisfy additional constraints that are potentially mentioned in the problem statement; 4. calling input() does not consume runtime nor memory, but of course any operations on top of it or afterwards will be counted towards runtime and memory footprint; 5. Anything printed gets added to the memory. You can take advantage of Python-specific optimizations provided by the underlying CPython interpreter or compiler to achieve the desired complexity, and you must account for them when analyzing the complexity.

Here is the programming question: context.

Here is the corresponding Python solution: code_content.

Please ignore any constraints on the input sizes that may have been previously mentioned in the problem description. Compute the big-Theta complexity as if inputs can be as large as possible.

Output the big-Theta time complexity only, no explanation needed, no other words needed.

### C.3    DETAILED BIG-O VARIANT - TIME COMPLEXITY PREDICTION

Provide the time complexity for the following competitive programming question and corresponding solution.

When analyzing the complexity of an algorithm, consider the worst-case scenario where all possible input combinations are tried, given the following conditions: 1. the inputs must adhere to the specified data types of the problem; 2. the inputs should not cause the code to crash or exit on an exception; 3. the inputs do not necessarily need to satisfy additional constraints that are potentially mentioned in the problem statement; 4. calling input() does not consume runtime nor memory, but of course any operations on top of it or afterwards will be counted towards runtime and memory footprint; 5. Anything printed gets added to the memory. You can take advantage of Python-specific optimizations provided by the underlying CPython interpreter or compiler to achieve the desired complexity, and you must account for them when analyzing the complexity.

Here is the programming question: context.

Here is the corresponding Python solution: code_content.

Please ignore any constraints on the input sizes that may have been previously mentioned in the problem description. Compute the big-O complexity as if inputs can be as large as possible.

The big-O complexity is the time complexity of the program when running on the worst case, which means you may want to find the worst running case first, and then find its corresponding time complexity, that you need to output in big-O format. For example, if the worst case is in the order of nlogn, do output precisely O(nlogn), as O(n**2) will be considered too high of an upper bound (though mathematically speaking one could say that what is O(nlogn) is also O(n**2)). Find the tightest possible upper bound in big-O notation.

Output the big-O time complexity only, no explanation needed, no other words needed.

## C.4 COT Big-O variant - Time Complexity Prediction

Provide the time complexity for the following competitive programming question and corresponding solution.

When analyzing the complexity of an algorithm, consider the worst-case scenario where all possible input combinations are tried, given the following conditions: 1. the inputs must adhere to the specified data types of the problem; 2. the inputs should not cause the code to crash or exit on an exception; 3. the inputs do not necessarily need to satisfy additional constraints that are potentially mentioned in the problem statement; 4. calling input() does not consume runtime nor memory, but of course any operations on top of it or afterwards will be counted towards runtime and memory footprint; 5. Anything printed gets added to the memory. You can take advantage of Python-specific optimizations provided by the underlying CPython interpreter or compiler to achieve the desired complexity, and you must account for them when analyzing the complexity.

Here is the programming question: context.

Here is the corresponding Python solution: code_content.

Please ignore any constraints on the input sizes that may have been previously mentioned in the problem description. Compute the big-O complexity as if inputs can be as large as possible.

Break down the analysis into these steps: 1. Identify all loops, recursive calls, and operations dependent on input size; 2. Explicitly state the time complexity of each Python-specific operation (e.g., list appends, dictionary lookups, sorting); 3. Assume all inputs are adversarially chosen to maximize runtime (e.g., hash collisions, worst-case comparisons); 4. Account for memory allocations (e.g., dynamic array resizes, string concatenation costs); 5. If recursion is used, include the cost of stack frames and potential tail-call optimizations; 6. Treat all arithmetic operations as O(1), but flag if arbitrary-precision integers could introduce hidden costs; 7. For nested operations, multiply complexities conservatively (e.g., O(n) loops inside O(n) loops = O(n²)); 8. Ignore Python's global interpreter lock (GIL) and concurrency effects; 9. Explicitly confirm whether built-in functions like sorted() or re.search() are treated as black-box with known complexities; 10. Final answer must be the tightest possible upper bound in big-O notation, even if the problem's original constraints imply smaller inputs.

Output the big-O time complexity at the end.

## C.5 ORIGINAL BIGOBENCH PROMPT - TIME COMPLEXITY RANKING

Provide a Python solution for the following competitive programming question: context.

Output the code only. Generate code that has an algorithmic time complexity of time_complexity. Try to optimize the runtime of your code as much as you can, while respecting the time complexity requirement.

When analyzing the complexity of an algorithm, consider the worst-case scenario where all possible input combinations are tried, given the following conditions: 1. the inputs must adhere to the specified data types of the problem; 2. the inputs should not cause the code to crash or exit on an exception; 3. the inputs do not necessarily need to satisfy additional constraints that are potentially mentioned in the problem statement; 4. calling input() does not consume runtime nor memory, but of course any operations on top of it or afterwards will be counted towards runtime and memory footprint; 5. Anything printed gets added to the memory. You can take advantage of Python-specific optimizations provided by the underlying CPython interpreter or compiler to achieve the desired complexity, and you must account for them when analyzing the complexity.

Your code should be enclosed in triple backticks like so: "'python YOUR CODE HERE "'. Use the backticks for your code only.

## C.6 PYTHON SPECIFICATIONS BIG-O VARIANT - TIME COMPLEXITY RANKING

Provide a Python solution for the following competitive programming question: context.

Output the code only. Generate code that has an algorithmic time complexity of time_complexity. Try to optimize the runtime of your code as much as you can, while respecting the time complexity requirement.

When analyzing the complexity of an algorithm, consider the worst-case scenario where all possible input combinations are tried, given the following conditions: 1. the inputs must adhere to the specified data types of the problem; 2. the inputs should not cause the code to crash or exit on an exception; 3. the inputs do not necessarily need to satisfy additional constraints that are potentially mentioned in the problem statement; 4. calling input() does not consume runtime nor memory, but of course any operations on top of it or afterwards will be counted towards runtime and memory footprint; 5. Anything printed gets added to the memory. You can take advantage of Python-specific optimizations provided by the underlying CPython interpreter or compiler to achieve the desired complexity, and you must account for them when analyzing the complexity.

As additional specifications: 1. Python Version - The code must be compatible with Python 3.10.0, avoid using features deprecated in this version or introduced in later versions, you can leverage the optimization that are proper to this version; 2. Libraries Allowed - You may use standard libraries (e.g., itertools, collections, math) and third-party libraries (e.g., NumPy, pandas) to optimize performance (For example, use numpy for vectorized operations on numerical data (e.g., np.where, np.concatenate), use collections.defaultdict for faster dictionary-like structures with default values...); 3. Memory Constraints - Avoid unnecessary copies of large data structures, Prefer in-place operations where possible (e.g., list.sort() instead of sorted() for large lists); 4. Edge Cases - Assume inputs adhere to the problem's data types and constraints (e.g., non-negative integers, valid strings) but do not need to handle invalid cases (e.g., non-integer inputs where integers are expected); 5. Output Handling - Minimize printing intermediate results to reduce memory overhead. Use generators or lazy evaluation (e.g., yield) for large datasets. Your code must still adhere strictly to the specified time complexity and avoid unnecessary operations.

Your code should be enclosed in triple backticks like so: "'python YOUR CODE HERE "'. Use the backticks for your code only.

## C.7 ABLATION RESULTS

Taking all these variations of the tasks' formulations in account, BIGO(BENCH) scores can be re-evaluated for each model. Table 5 compares prompt results across complexity prediction and generation, while Table 6 look at the complexity ranking task. For all tasks, only the time complexity version was tried out.

As seen on these figures, the reasoning model's performance on Time Complexity Prediction decreased slightly with the big-Theta and detailed big-O variants, and significantly with the COT-prompted variant, reaching an All@1 score of 31.8 (compared to the official BigOBench score of 40.4). The non-reasoning model's performance also decreased slightly with the big-Theta variant, but improved with the detailed big-O and COT variants, reaching 36.1 All@1. We believe that sharing more details with the reasoning model leads to overthinking, while the non-reasoning model benefits from the added context. The non-reasoning model's performance, even with COT, remains significantly lower than the reasoning model's.

In the case of Time Complexity Generation, the non-reasoning model falls short of seeing any improvements with the different prompt variants. We believe the performance is so low (3.3 All@1 for Llama 3.3 70B) that even sharing slightly more details can not help a model that simply does not grasp the objective of the task. In the case of the reasoning model, we do see an improvement using the variants, up to scores of 11.7 All@1 with the big-Theta variant. The base reported score of Qwen QwQ on this task is 9.6 All@1, which in our intuition means the model starts to grasp the task, though it lacks guidance compared to the Complexity Prediction equivalent (40.4 All@1). In this case, we believe the added details to the prompt helps the model perform better while staying far from overthinking. It also helps put more focus on the complexity requirement compared to code correctness, given that a manual error analysis underlined the models often lose focus from the first objective during the reasoning process.

Finally, adding python specifications to the prompt resulted in a marginal improvement in performance (up to +2 coeffFULL for Qwen QwQ).

Table 5: BIGO(BENCH) benchmark results for variation of the task prompts on a reasoning model (QWEN QWQ 32B) and a non-reasoning model (LLAMA 3.3 70B). **Program Synthesis** checks the correctness of model-generated solutions to given programming problems, not taking into account any complexity requirement. **Complexity Prediction** measures whether a model can find the time-space complexity of an existing code snippet. **Complexity Generation** evaluates whether a model can output a working code snippet to a given problem that meets a time-space complexity requirement. $Pass@k$ considers each complexity class of all problems independently and calculates a macro-average between them. $Best@k$ is a refinement of $Pass@k$, focusing only on the most optimized complexity class for each problem. $All@k$ checks if all complexity classes for each problem are correctly predicted or generated simultaneously, then macro-averages across all problems.

| | PROG. SYNTHESIS | | COMPLEXITY PRED. | | | COMPLEXITY GEN. | | | |
| --- | --- | --- | --- | --- | --- | --- | --- | --- | --- |
| MODEL | PASS @1 | PASS @10 | PASS @1 | BEST @1 | ALL @1 | PASS @1 | PASS @10 | BEST @1 | ALL @1 |
| **QWEN QWQ** | | | | | | | | | |
| BIGOBENCH PROMPT | 60.7 | 75.8 | 62.7 | 72.2 | 40.3 | 26.9 | 50.9 | 37.8 | 8.3 |
| BIG THETA | 58.4 | 74.8 | 62.0 | 71.4 | 39.9 | 28.8 | 52.1 | 37.3 | 11.7 |
| BIGO WITH MORE DETAILS | 58.4 | 74.7 | 61.9 | 70.8 | 39.5 | 26.7 | 51.3 | 36.7 | 8.8 |
| BIGO WITH COT | 57.9 | 74.1 | 57.1 | 62.9 | 31.8 | 27.0 | 51.6 | 36.3 | 9.7 |
| **LLAMA 3.3** | | | | | | | | | |
| BIGOBENCH PROMPT | 34.0 | 54.9 | 58.0 | 72.3 | 33.7 | 13.1 | 31.3 | 19.1 | 2.1 |
| BIG THETA | 33.5 | 54.8 | 57.2 | 72.7 | 32.1 | 12.3 | 30.8 | 18.5 | 1.8 |
| BIGO WITH MORE DETAILS | 33.5 | 54.5 | 59.9 | 73.6 | 35.5 | 12.5 | 31.1 | 18.3 | 1.9 |
| BIGO WITH COT | 37.0 | 62.4 | 59.3 | 68.1 | 36.1 | 14.1 | 37.5 | 20.3 | 2.2 |

Table 6: BIGO(BENCH) benchmark results for variation of the task prompts on a reasoning model (QWEN QwQ 32B) and a non-reasoning model (LLAMA 3.3 70B). **Program Synthesis** checks the correctness of model-generated solutions to given programming problems, not taking into account any complexity requirement. **Complexity Ranking** evaluates whether a model can output a working code snippet to a given problem that meets a time-space complexity requirement, while being as optimized as possible within this complexity class of solutions. $Pass@k$ considers each complexity class of all problems independently and calculates a macro-average between them. $Best@k$ is a refinement of $Pass@k$, focusing only on the most optimized complexity class for each problem. $All@k$ checks if all complexity classes for each problem are correctly predicted or generated simultaneously, then macro-averages across all problems. Finally for $CoeffFull$, using the complexity framework, the best measured coefficient of the complexity curve, out of 20 attempts, is used to rank LLM-generated code among human solutions from the same problem and time-space complexity class. Ranking is percentile based, n% ranking score amounts for n% human solutions having worse complexity coefficient. If no LLM solution passes correctness tests, ranking score is set to 0.

| | PROGRAM SYNTHESIS | | COMPLEXITY RANKING | | | | |
|---|---|---|---|---|---|---|---|
| MODEL | PASS @1 | PASS @10 | PASS @1 | PASS @10 | BEST @1 | ALL @1 | COEFFFULL |
| **QWEN QwQ** | | | | | | | |
| BIGOBENCH PROMPT | 68.8 | 82.8 | 27.7 | 54.8 | 40.6 | 6.9 | 48.1 |
| BIGO WITH PYTHON DETAILS | 68.9 | 82.5 | 26.6 | 55.3 | 37.7 | 6.6 | 49.8 |
| **LLAMA 3.3** | | | | | | | |
| BIGOBENCH PROMPT | 36.3 | 58.8 | 13.6 | 33.4 | 20.1 | 2.1 | 29.5 |
| BIGO WITH PYTHON DETAILS | 33.8 | 57.0 | 10.9 | 32.8 | 16.4 | 1.4 | 30.2 |

## D  MULTI-SAMPLE RESULTS SCORES

For any of the metrics defined for the tasks of **Complexity Prediction**, **Complexity Generation** and **Complexity Ranking**, the models are queried multiple times in order to provide accurate unbiased estimators for each of those metrics. In the specific case of **Complexity Prediction**, we do only report @1 scores: this is because this task remains a classification and not a generative task, having an output space small and fixed, in which case repeated sampling would conflate model uncertainty with true performance, instead of reflecting some true exploration capabilities such as in code generation.

As defined in Section 5 and in Section A, we define **Pass@k** for Complexity Generation for example following the definition of Chen et al. (2021a):

$$
\text{Pass@k} = \mathbb{E}_{P \sim \mathcal{P}} \left[ \frac{1}{|\mathcal{C}(P)|} \sum_{c \in \mathcal{C}(P)} \left( 1 - \frac{\binom{n - c_c}{k}}{\binom{n}{k}} \right) \right],
$$

where $c_c = \sum_{i=1}^{n} \mathbb{I}_{\text{class}}(S'_i, c, P) \times \mathbb{I}_{\text{correct}}(S'_i, P)$.

Table 7: BIGO(BENCH) benchmark results for popular LLMs on the tasks of **Program Synthesis** and **Complexity Generation**, with multiple @$k$ estimators. **Program Synthesis** checks the correctness of model-generated solutions to given programming problems, not taking into account any complexity requirement. **Complexity Generation** evaluates whether a model can output a working code snippet to a given problem that meets a time-space complexity requirement. $Pass@k$ considers each complexity class of all problems independently and calculates a macro-average between them. $Best@k$ is a refinement of $Pass@k$, focusing only on the most optimized complexity class for each problem. $All@k$ checks if all complexity classes for each problem are correctly predicted or generated simultaneously, then macro-averages across all problems.

| MODEL | PROG. SYNTHESIS | | | | | | COMPLEXITY GEN. | | | | | |
|---|---|---|---|---|---|---|---|---|---|---|---|---|
| | PASS @1 | PASS @5 | PASS @10 | PASS @1 | PASS @5 | PASS @10 | BEST @1 | BEST @5 | BEST @10 | ALL @1 | ALL @5 | ALL @10 |
| **TIME** | | | | | | | | | | | | |
| BASELINES | 30.3 | 48.5 | 55.4 | 12.1 | 23.7 | 29.7 | 19.0 | 33.5 | 40.3 | 0.9 | 3.7 | 6.0 |
| LLAMA 3.3 70B | 43.4 | 60.7 | 66.0 | 17.7 | 32.9 | 40.0 | 25.7 | 41.8 | 47.8 | 3.3 | 9.5 | 13.9 |
| LLAMA 4 SCOUT 17BX16E | 61.7 | 76.9 | 80.1 | 22.8 | 40.5 | 48.0 | 31.8 | 50.8 | 58.2 | 3.5 | 10.7 | 16.2 |
| LLAMA 4 MAVERICK 17BX128E | 59.3 | 74.6 | 78.9 | 19.9 | 37.0 | 44.6 | 27.0 | 45.9 | 52.4 | 5.3 | 12.1 | 17.1 |
| GEMMA 3 27B | 37.7 | 43.7 | 45.6 | 15.2 | 19.3 | 20.8 | 17.6 | 21.9 | 23.4 | 1.8 | 3.2 | 3.9 |
| CODESTRAL 22B | 23.7 | 40.5 | 47.5 | 10.6 | 21.2 | 26.6 | 14.9 | 27.6 | 33.8 | 1.3 | 4.3 | 6.4 |
| QWEN2.5-CODER 32B | 30.5 | 45.3 | 50.8 | 12.2 | 22.0 | 26.5 | 15.2 | 26.8 | 31.5 | 3.1 | 6.6 | 8.7 |
| GPT-4O | 51.0 | 73.1 | 78.3 | 20.6 | 37.5 | 44.7 | 30.2 | 51.9 | 59.5 | 4.3 | 11.7 | 16.4 |
| DEEPSEEKCODERV2 236B | 44.1 | 60.5 | 65.5 | 19.5 | 32.3 | 38.0 | 27.6 | 43.0 | 48.7 | 3.3 | 8.1 | 11.8 |
| DEEPSEEKV3 671B | 41.4 | 58.4 | 63.6 | 17.7 | 31.5 | 37.7 | 23.0 | 39.6 | 46.2 | 3.4 | 8.9 | 12.2 |
| DEEPSEEKR1 QWEN 32B | 70.1 | 81.2 | 83.7 | 29.0 | 44.1 | 49.9 | 46.1 | 61.7 | 66.0 | 4.8 | 14.7 | 21.0 |
| DEEPSEEKR1 LLAMA 70B | 70.1 | 81.3 | 83.8 | 29.2 | 45.3 | 51.6 | 46.5 | 63.4 | 68.4 | 4.8 | 15.5 | 22.6 |
| LLAMA 3.1 NEMOTRON-ULTRA 253B | 80.0 | 88.1 | 89.4 | 33.5 | 49.0 | 54.9 | 51.8 | 66.5 | 71.5 | 6.1 | 16.8 | 23.2 |
| QWEN3 32B | 70.0 | 79.4 | 81.3 | 29.1 | 47.2 | 53.8 | 43.5 | 61.9 | 67.2 | 6.5 | 19.0 | 26.1 |
| **SPACE** | | | | | | | | | | | | |
| BASELINES | 30.1 | 46.8 | 52.6 | 12.2 | 25.3 | 32.4 | 17.8 | 32.7 | 40.0 | 1.3 | 5.4 | 8.9 |
| LLAMA 3.3 70B | 42.6 | 58.1 | 62.5 | 15.0 | 30.6 | 37.7 | 21.9 | 38.7 | 45.2 | 1.8 | 6.2 | 10.0 |
| LLAMA 4 SCOUT 17BX16E | 56.7 | 70.3 | 73.5 | 20.0 | 34.3 | 40.1 | 31.1 | 47.9 | 53.9 | 2.2 | 8.2 | 12.8 |
| LLAMA 4 MAVERICK 17BX128E | 58.4 | 72.0 | 75.7 | 16.8 | 24.7 | 28.2 | 30.4 | 41.2 | 45.2 | 0.8 | 3.0 | 4.6 |
| GEMMA 3 27B | 40.3 | 46.9 | 49.0 | 16.2 | 22.1 | 24.3 | 22.5 | 29.8 | 31.9 | 1.4 | 3.5 | 5.1 |
| CODESTRAL 22B | 25.7 | 41.4 | 47.6 | 11.0 | 23.3 | 29.4 | 16.7 | 31.1 | 37.1 | 1.3 | 5.3 | 8.8 |
| QWEN2.5-CODER 32B | 31.1 | 44.6 | 49.2 | 10.1 | 19.2 | 23.3 | 15.3 | 26.4 | 30.5 | 1.2 | 4.0 | 6.2 |
| GPT-4O | 51.6 | 70.8 | 74.4 | 18.1 | 33.7 | 39.9 | 28.0 | 47.0 | 53.0 | 1.4 | 6.0 | 10.1 |
| DEEPSEEKCODERV2 236B | 43.1 | 58.8 | 63.8 | 16.7 | 29.0 | 34.5 | 25.6 | 40.0 | 45.0 | 1.0 | 4.1 | 7.0 |
| DEEPSEEKV3 671B | 41.8 | 57.9 | 62.5 | 15.0 | 29.1 | 35.4 | 22.6 | 40.4 | 46.8 | 1.6 | 5.6 | 8.8 |
| DEEPSEEKR1 QWEN 32B | 68.0 | 78.7 | 80.6 | 24.8 | 41.9 | 48.6 | 38.6 | 58.7 | 64.3 | 3.1 | 11.4 | 17.8 |
| DEEPSEEKR1 LLAMA 70B | 68.8 | 79.3 | 81.2 | 25.6 | 43.4 | 50.0 | 38.7 | 59.1 | 64.6 | 3.4 | 12.2 | 18.8 |
| LLAMA 3.1 NEMOTRON-ULTRA 253B | 77.7 | 85.2 | 86.4 | 30.4 | 48.7 | 55.5 | 45.3 | 65.6 | 70.9 | 5.6 | 16.4 | 23.7 |
| QWEN3 32B | 65.9 | 75.5 | 77.7 | 25.5 | 41.9 | 47.8 | 39.6 | 58.7 | 64.2 | 5.1 | 14.0 | 19.5 |

The main results are summarized in Table 2 which includes pass@10 scores for program synthesis and complexity generation scores. On top of it, we present in Table 7 more details on performance when the generation budget allows for multiple attempts (note that for O1-MINI, due to limited inference budget, we could not retrieve the scores at higher @$k$ values). For **Program Synthesis**, we include $pass@\{1, 5, 10\}$ scores, along with $best@\{1, 5, 10\}$ and $all@\{1, 5, 10\}$ scores for **Complexity Generation** and **Complexity Ranking**.

Table 8: BIGO(BENCH) benchmark results for popular LLMs on the tasks of **Complexity Generation**, with multiple @$k$ estimators, where @5 and @10 are displayed relative to the corresponding @1 score. **Complexity Generation** evaluates whether a model can output a working code snippet to a given problem that meets a time-space complexity requirement. $Pass@k$ considers each complexity class of all problems independently and calculates a macro-average between them. $All@k$ checks if all complexity classes for each problem are correctly predicted or generated simultaneously, then macro-averages across all problems.

| MODEL | COMPLEXITY GEN. | | | | | |
| | PASS @1 | PASS @5 | PASS @10 | ALL @1 | ALL @5 | ALL @10 |
|---|---|---|---|---|---|---|
| **TIME** | | | | | | |
| LLAMA 3.3 70B | 17.7 | +85.8% | +126.2% | 3.3 | +183.4% | +315.0% |
| LLAMA 4 SCOUT 17BX16E | 22.8 | +77.4% | +110.4% | 3.5 | +206.8% | +365.2% |
| LLAMA 4 MAVERICK 17BX128E | 19.9 | +86.4% | +124.8% | 5.3 | +127.9% | +221.0% |
| GEMMA 3 27B | 15.2 | +27.1% | +37.5% | 1.8 | +81.7% | +122.6% |
| CODESTRAL 22B | 10.6 | +100.5% | +151.6% | 1.3 | +222.8% | +382.2% |
| QWEN2.5-CODER 32B | 12.2 | +80.0% | +116.6% | 3.1 | +114.1% | +182.4% |
| GPT-4O | 20.6 | +81.5% | +116.4% | 4.3 | +170.8% | +280.0% |
| DEEPSEEKCODERV2 236B | 19.5 | +66.0% | +95.0% | 3.3 | +141.1% | +252.6% |
| DEEPSEEKV3 671B | 17.7 | +78.2% | +113.3% | 3.4 | +157.4% | +254.9% |
| DEEPSEEKR1 QWEN 32B | 29.0 | +52.1% | +72.1% | 4.8 | +202.8% | +333.3% |
| DEEPSEEKR1 LLAMA 70B | 29.2 | +55.2% | +76.8% | 4.8 | +223.2% | +371.3% |
| LLAMA 3.1 NEMOTRON-ULTRA 253B | 33.5 | +46.4% | +63.8% | 6.1 | +175.1% | +279.8% |
| QWEN3 32B | 29.1 | +62.0% | +84.7% | 6.5 | +190.7% | +299.7% |
| **SPACE** | | | | | | |
| LLAMA 3.3 70B | 15.0 | +103.9% | +151.1% | 1.8 | +256.0% | +471.5% |
| LLAMA 4 SCOUT 17BX16E | 20.0 | +71.4% | +100.5% | 2.2 | +279.3% | +490.7% |
| LLAMA 4 MAVERICK 17BX128E | 16.8 | +47.2% | +68.5% | 0.8 | +266.9% | +471.5% |
| GEMMA 3 27B | 16.2 | +36.5% | +50.1% | 1.4 | +145.5% | +253.5% |
| CODESTRAL 22B | 11.0 | +111.5% | +166.7% | 1.3 | +319.6% | +592.4% |
| QWEN2.5-CODER 32B | 10.1 | +90.3% | +131.3% | 1.2 | +245.5% | +434.2% |
| GPT-4O | 18.1 | +86.3% | +120.5% | 1.4 | +318.7% | +608.4% |
| DEEPSEEKCODERV2 236B | 16.7 | +73.1% | +106.1% | 1.0 | +298.8% | +572.4% |
| DEEPSEEKV3 671B | 15.0 | +93.4% | +135.3% | 1.6 | +256.8% | +459.8% |
| DEEPSEEKR1 QWEN 32B | 24.8 | +68.7% | +95.6% | 3.1 | +264.3% | +470.9% |
| DEEPSEEKR1 LLAMA 70B | 25.6 | +69.8% | +95.3% | 3.4 | +259.0% | +452.6% |
| LLAMA 3.1 NEMOTRON-ULTRA 253B | 30.4 | +60.4% | +82.6% | 5.6 | +192.3% | +322.7% |
| QWEN3 32B | 25.5 | +63.9% | +87.2% | 5.1 | +176.3% | +286.0% |

When pushing all@k scores to k=10, the new best scores are obtained by Qwen3 32B on time complexity generation with 26.1 all@10 (also best model for all@1 with 6.5) and Llama 3.1 Nemotron-Ultra 253B on space complexity generation with 23.7 all@10 (also best model for all@1 with 5.6).

In general, the order of models does not change much when pushing @k scores to higher values, except for Gemma 3, which shows less performance return. This is especially visible in Fig. 6, which provides the score evolution for all models across 1-step increments of k on time and space complexity tasks. Reasoning models benefit more from higher @k generations than non-reasoning models, especially on time complexity generation. Across all models, we notice higher gains for higher @k metrics when using all@k, and on space complexity generation.

Across all models, we notice higher gains for higher @k metrics when using all@k (compared to pass@k), and on space complexity generation (compared to time complexity generation). On average, models get +100% performance for time pass@1 to pass@10, +275% time all@1 to all@10, and on space complexity generation respectively +110% and up to +450%. In general, the more challenging the tasks, the higher the gains for higher @k values.

Finally, for time complexity generation, we observe more marginal gains the further we improve @k values on all@k scores, compared to pass@k scores. For example, on average across models, pass@5 is 70% higher than pass@1, whereas pass@10 is only 15% higher than pass@5. In comparison, all@5 is a staggering 160% higher than all@1, and all@10 remains 40% higher than all@5. Looking at the difference between @10 and @9, all@k still grows two times faster than pass@k.

This gives more extensive details on the strategies to adopt to use the compute budget where it matters the most in the context of complexity related tasks. Though some tasks remain very challenging

at the single-sample level (all models on time-space complexity generation keep all@1 score below 10), even just doubling the sampling budget already leads to substantial performance gains.

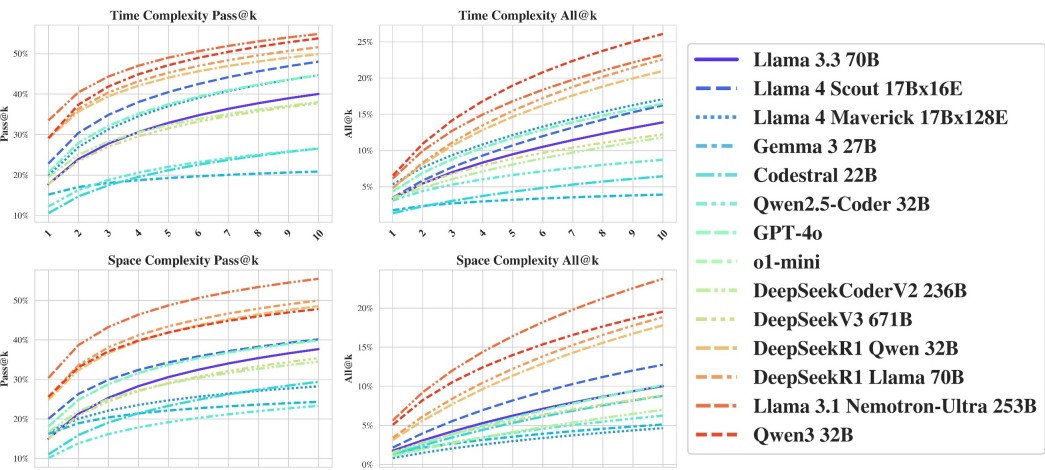

Figure 6: Comparing gains of $Pass@k$ and $All@k$ across all models on the task of **Complexity Generation**.

# E    FINE-TUNING EXPERIMENTS MORE RESULTS

In this section, we provide more extensive results about the SFT experiments. Table 9 displays multiple @k values for the LLAMA 3.1 70B model queried for zero-shot, few-shot, and then further trained with either **Time Prediction Fine-tuning**, **Space Prediction Fine-tuning**, **Time Generation Fine-tuning**, or **Space Generation Fine-tuning**, evaluated for the **Time Complexity** tasks. Similarly, Table 11 displays the same methods but evaluated on **Space Complexity** tasks.

Table 10 provides relative scores for @5 and @10 on **Time Complexity**, and Table 12 for **Space Complexity**.

Finally, Fig. 7 displays visually the gain on the task of **Complexity Generation**.

Table 9: Time Complexity BIGO(BENCH) benchmark results when fine-tuning LLAMA 3.1 70B on the tasks of **Time Complexity Prediction**, **Space Complexity Prediction**, **Time Complexity Generation** and **Space Complexity Generation**, then evaluated on the tasks of **Program Synthesis** and **Time Complexity Generation**, with multiple @k estimators. **Program Synthesis** checks the correctness of model-generated solutions to given programming problems, not taking into account any complexity requirement. **Complexity Generation** evaluates whether a model can output a working code snippet to a given problem that meets a time-space complexity requirement. $Pass@k$ considers each complexity class of all problems independently and calculates a macro-average between them. $Best@k$ is a refinement of $Pass@k$, focusing only on the most optimized complexity class for each problem. $All@k$ checks if all complexity classes for each problem are correctly predicted or generated simultaneously, then macro-averages across all problems.

| METHOD | PROG. SYNTHESIS | | | TIME COMPLEXITY GEN. | | | | | | | | |
| | PASS @1 | PASS @5 | PASS @10 | PASS @1 | PASS @5 | PASS @10 | BEST @1 | BEST @5 | BEST @10 | ALL @1 | ALL @5 | ALL @10 |
|---|---|---|---|---|---|---|---|---|---|---|---|---|
| ZERO-SHOT | 29.6 | 47.2 | 54.0 | 14.2 | 27.9 | 34.8 | 20.3 | 35.0 | 41.2 | 3.1 | 8.9 | 13.0 |
| FEW-SHOT | 28.9 | 45.9 | 52.8 | 13.4 | 26.4 | 33.0 | 19.6 | 35.0 | 41.8 | 2.4 | 6.6 | 9.6 |
| **PREDICTION FINE-TUNING** | | | | | | | | | | | | |
| ON TIME | 27.4 | 46.4 | 53.6 | 12.6 | 27.2 | 34.2 | 17.2 | 32.6 | 39.0 | 2.9 | 9.1 | 13.2 |
| ON SPACE | 26.6 | 46.4 | 54.3 | 12.3 | 26.6 | 34.2 | 17.3 | 33.5 | 40.9 | 2.4 | 7.6 | 11.3 |
| **GENERATION FINE-TUNING** | | | | | | | | | | | | |
| ON TIME | 23.2 | 42.0 | 48.5 | 10.0 | 23.1 | 29.6 | 13.3 | 27.6 | 33.6 | 1.2 | 4.5 | 7.0 |
| ON SPACE 17Bx128E | 23.4 | 42.7 | 50.0 | 9.9 | 23.0 | 29.8 | 13.1 | 27.8 | 34.9 | 1.5 | 5.6 | 8.7 |

Table 10: Time Complexity BIGO(BENCH) benchmark results when fine-tuning LLAMA 3.1 70B on the tasks of **Time Complexity Prediction**, **Space Complexity Prediction**, **Time Complexity Generation** and **Space Complexity Generation**, then evaluated on the tasks of **Time Complexity Generation**, with multiple @k estimators, where @5 and @10 are displayed relative to the corresponding @1 score. **Complexity Generation** evaluates whether a model can output a working code snippet to a given problem that meets a time-space complexity requirement. $Pass@k$ considers each complexity class of all problems independently and calculates a macro-average between them. $All@k$ checks if all complexity classes for each problem are correctly predicted or generated simultaneously, then macro-averages across all problems.

| MODEL | TIME COMPLEXITY GEN. | | | | | |
| | PASS @1 | PASS @5 | PASS @10 | ALL @1 | ALL @5 | ALL @10 |
|---|---|---|---|---|---|---|
| ZERO-SHOT | 14.2 | +95.9% | +144.2% | 3.1 | +188.1% | +322.1% |
| FEW-SHOT | 13.4 | +97.0% | +145.6% | 2.4 | +173.4% | +293.4% |
| **PREDICTION FINE-TUNING** | | | | | | |
| ON TIME | 12.6 | +115.6% | +170.8% | 2.9 | +215.8% | +359.6% |
| ON SPACE | 12.3 | +116.9% | +178.6% | 2.4 | +216.0% | +371.6% |
| **GENERATION FINE-TUNING** | | | | | | |
| ON TIME | 10.0 | +131.4% | +196.3% | 1.2 | +271.3% | +481.0% |
| ON SPACE | 9.9 | +132.1% | +200.8% | 1.5 | +260.8% | +464.4% |

Table 11: Space Complexity BIGO(BENCH) benchmark results when fine-tuning LLAMA 3.1 70B on the tasks of **Time Complexity Prediction**, **Space Complexity Prediction**, **Time Complexity Generation** and **Space Complexity Generation**, then evaluated on the tasks of **Program Synthesis** and **Space Complexity Generation**, with multiple $@k$ estimators. **Program Synthesis** checks the correctness of model-generated solutions to given programming problems, not taking into account any complexity requirement. **Complexity Generation** evaluates whether a model can output a working code snippet to a given problem that meets a time-space complexity requirement. $Pass@k$ considers each complexity class of all problems independently and calculates a macro-average between them. $Best@k$ is a refinement of $Pass@k$, focusing only on the most optimized complexity class for each problem. $All@k$ checks if all complexity classes for each problem are correctly predicted or generated simultaneously, then macro-averages across all problems.

| | PROG. SYNTHESIS | | | SPACE COMPLEXITY GEN. | | | | | | | | |
|---|---|---|---|---|---|---|---|---|---|---|---|---|
| METHOD | PASS @1 | PASS @5 | PASS @10 | PASS @1 | PASS @5 | PASS @10 | BEST @1 | BEST @5 | BEST @10 | ALL @1 | ALL @5 | ALL @10 |
| ZERO-SHOT | 29.7 | 45.7 | 52.3 | 11.7 | 25.5 | 33.0 | 17.2 | 32.4 | 39.1 | 1.8 | 6.9 | 10.9 |
| FEW-SHOT | 29.8 | 46.5 | 53.3 | 11.3 | 24.8 | 31.7 | 16.4 | 31.6 | 38.6 | 1.4 | 5.5 | 8.9 |
| PREDICTION FINE-TUNING | | | | | | | | | | | | |
| ON TIME | 27.3 | 44.7 | 51.4 | 10.5 | 24.5 | 32.1 | 15.0 | 30.2 | 36.8 | 1.3 | 5.6 | 9.6 |
| ON SPACE | 27.0 | 45.0 | 51.9 | 10.5 | 24.4 | 31.8 | 15.2 | 30.9 | 38.1 | 1.4 | 5.6 | 9.0 |
| GENERATION FINE-TUNING | | | | | | | | | | | | |
| ON TIME | 23.9 | 42.6 | 49.5 | 9.9 | 23.9 | 31.4 | 14.6 | 31.1 | 38.3 | 1.3 | 5.7 | 9.5 |
| ON SPACE 17Bx128E | 24.2 | 42.1 | 48.8 | 10.3 | 23.7 | 30.4 | 15.0 | 31.1 | 37.6 | 1.4 | 5.3 | 8.8 |

Table 12: Space Complexity BIGO(BENCH) benchmark results when fine-tuning LLAMA 3.1 70B on the tasks of **Time Complexity Prediction**, **Space Complexity Prediction**, **Time Complexity Generation** and **Space Complexity Generation**, then evaluated on the tasks of **Space Complexity Generation**, with multiple $@k$ estimators, where @5 and @10 are displayed relative to the corresponding @1 score. **Complexity Generation** evaluates whether a model can output a working code snippet to a given problem that meets a time-space complexity requirement. $Pass@k$ considers each complexity class of all problems independently and calculates a macro-average between them. $All@k$ checks if all complexity classes for each problem are correctly predicted or generated simultaneously, then macro-averages across all problems.

| | SPACE COMPLEXITY GEN. | | | | | |
|---|---|---|---|---|---|---|
| MODEL | PASS @1 | PASS @5 | PASS @10 | ALL @1 | ALL @5 | ALL @10 |
| ZERO-SHOT | 11.7 | +117.3% | +181.5% | 1.8 | +284.0% | +509.9% |
| FEW-SHOT | 11.3 | +119.3% | +180.5% | 1.4 | +291.1% | +538.4% |
| PREDICTION FINE-TUNING | | | | | | |
| ON TIME | 10.5 | +133.8% | +206.0% | 1.3 | +332.0% | +638.9% |
| ON SPACE | 10.5 | +133.2% | +203.4% | 1.4 | +298.3% | +533.7% |
| GENERATION FINE-TUNING | | | | | | |
| ON TIME | 9.9 | +141.2% | +217.7% | 1.3 | +324.9% | +614.1% |
| ON SPACE | 10.3 | +131.2% | +196.4% | 1.4 | +289.1% | +548.8% |

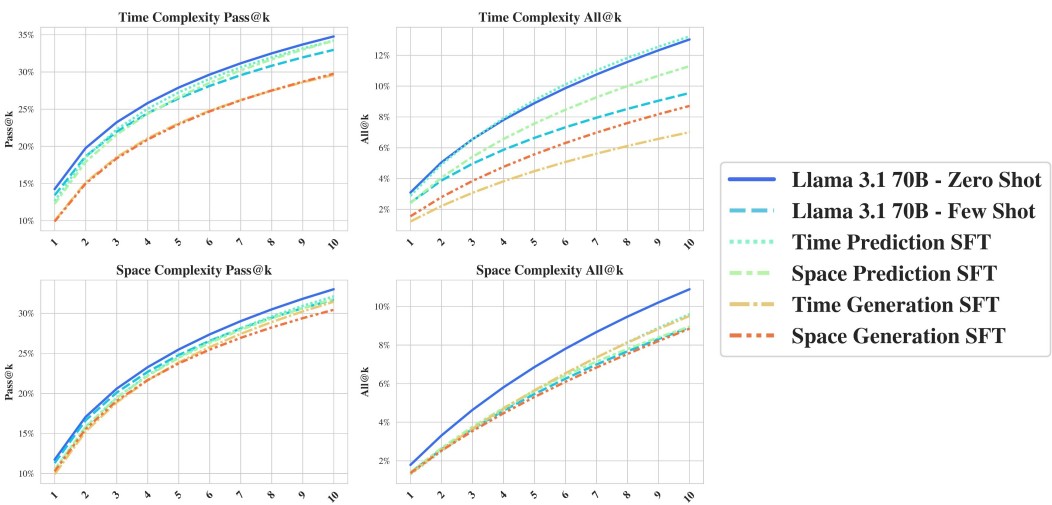

Figure 7: Comparing gains of $Pass@k$ and $All@k$ across LLAMA 3.1 70B variants on the task of **Complexity Generation**.

# F  TASKS CORRELATIONS

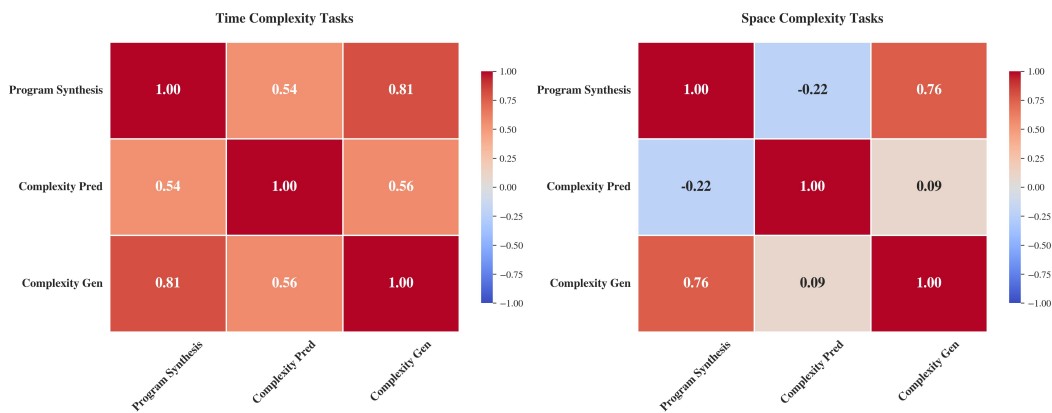

Figure 8: Correlation matrices comparing the distribution of scores across all models benchmarked on BIGO(BENCH) for the tasks of **Program Synthesis** (on the corresponding time or space complexity test set), **Complexity Prediction** and **Complexity Generation**, the two latter being respectfully on the time and space complexity test sets. Correlations are computed with the Pearson standard correlation coefficient.

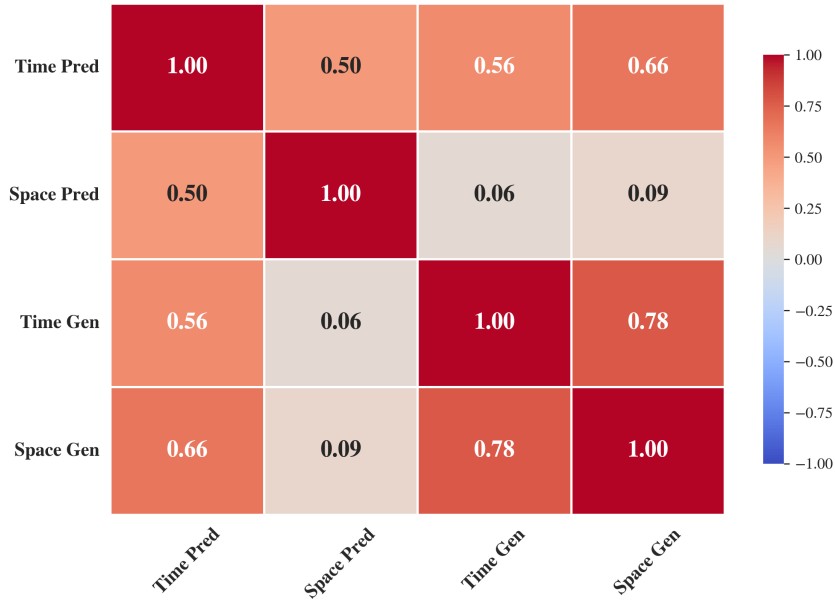

Figure 9: Correlation matrix comparing the distribution of scores across all models benchmarked on BIGO(BENCH) for the tasks of **Time Complexity Prediction** and **Time Complexity Generation**, along **Space Complexity Prediction** and **Space Complexity Generation**. Correlations are computed with the Pearson standard correlation coefficient.

Leveraging the many scores computed across models on the different tasks of BIGO(BENCH), correlations can be measured between scores and tasks (program synthesis, complexity prediction and complexity generation for both time and space). These are depicted in Fig. 8 for correlations matrices within time-related tasks and within space-related tasks. On top of that, Fig. 9 measures

correlations between time and space prediction and generation tasks. All of these measures are based on Pearson standard correlation coefficient (Pearson & Galton, 1895).

Overall, the correlations are the highest between program synthesis and complexity generation, respectively 0.81 and 0.76 for time and space complexity, across all models. Intuitively, we believe that this is because models are more frequently exposed to the time optimization objective when being trained for generating code, as this objective is more popular (on code competition platforms for example) and more documented than its space equivalent. Time prediction has a correlation coefficient of 0.56 with time complexity generation, meaning that the program synthesis objective may dominate on this task, when the model is trying to answer this double-requirement task. On space generation, a task where we see very low all@1 scores, the correlation between prediction and generation falls down to 0.1. This is probably explained by the very low performance on space generation, models being confused by the notion of generation under a space complexity constraint, therefore losing focus on this objective and mostly correlating with the performance on more simple program synthesis.

Finally, when comparing time and space tasks, we observe that time prediction correlates more with time generation than with space prediction, but that the converse does not hold, as space prediction has a correlation coefficient of 0.5 with time prediction and only 0.1 with space generation, a task that is dominated by the program synthesis objective, given that the double objective remains too hard for most models. Time generation and space generation both correlate the most with one another, compared to their respective prediction counterparts.

Notice that the fine-tuning experiments presented Table 4 and discussed in Section 6 can also help better understand the dependencies between the tasks introduced in BIGO(BENCH).

## G    FRAMEWORK IMPLEMENTATION DETAILS

### G.1    GENERAL FRAMEWORK OUTLINE

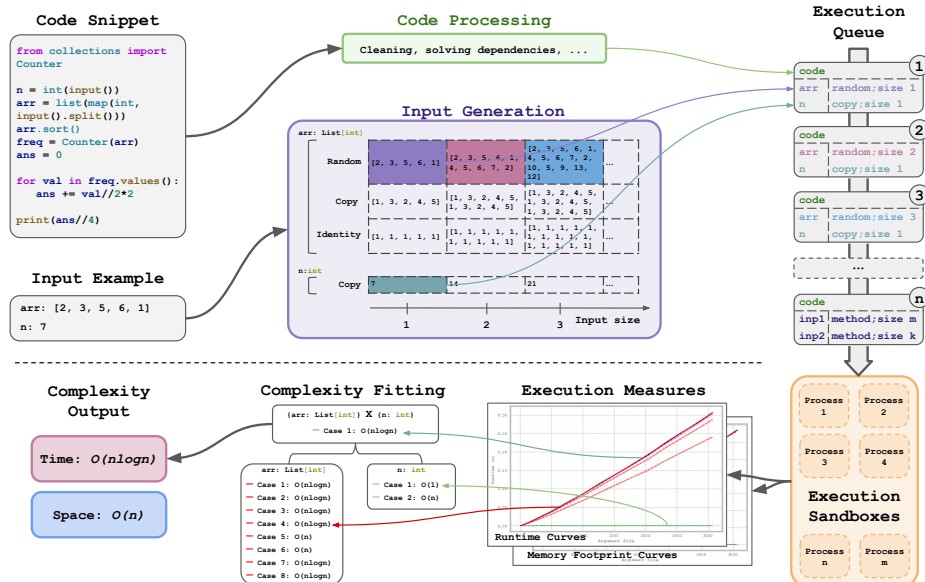

Figure 10: Outline of the dynamic complexity inference framework. The framework takes a code snippet and a single example of inputs to this code snippet. Then, it processes the code snippet and proceeds with extensive inputs generation, based on the provided example of inputs: inputs are independently or interdependently increased in size, using several expansion methods that can be the identity or random, among else. This forms a queue of synthetic inputs on which to execute the provided code snippet. These executions happen independently in sandboxes, where runtime and memory footprint measures are taken. Once all the measures are collected, the framework can model the code snippet time and space dependencies to the different inputs. Using curve fitting, the time and space complexity of the code is computed on each input separately and then altogether. The global time and space complexity over all inputs is what is being returned.

The time-space complexity framework is a rule-based algorithm that can process any Python function in order to infer its time and space complexities dynamically. The high-level principles of the framework are presented in Fig. 10, explaining how it takes a code-snippet and an input example and try to infer a time and space complexity from it.

As inputs, it takes a Python function along its function inputs and their corresponding dataclass, which are then processed and modified before being run while runtime and memory footprints are measured. From a high-level perspective, the framework increases the size of inputs following various strategies, in order to assess the impact of their size on execution metrics (e.g. execution time, memory used). When the function has several arguments, they can be expanded independently or together to determine the overall complexity of the function, taking into account potential interdependencies. The prepared code, along with the various sets of expanded inputs are queued up and run in independent sandboxes, using the *Bubblewrap* library (bub, 2024), to avoid any harmful side effects of the code being run. While running, *Cprofiler* (cpy, 2024) is used for time execution measures and *tracemalloc* for memory footprint. Using non-negative least squares curve fitting on each set of measures, the coefficients and residuals of each complexity class are computed. The gold complexity class output for a given set of measures is chosen as the minimizer of the residuals, taking into account a simplicity bias (the more simple the complexity class is, the smaller the simplicity bias). This curve fitting is applied on each set of measures, each corresponding to a different subset of arguments being expanded with a different expansion method. Using ensemble methods, the global complexity of the Python function is computed by aggregating the individual complexity outputs along the different set of measures. Finally, the complexity framework also returns the co-

efficients of the curve of each elected complexity. These coefficients can be leveraged to rank and classify the optimizations of different Python solutions within the same complexity class.

## G.2 COMPLEXITY FRAMEWORK INPUTS

The complexity framework handles two types on input codes, on which to measure time and space complexity. The released datasets (as part of our project on HuggingFace), as well as all the results detailed in our paper, do follow format 1, but in case you need it for different input data, the framework can also handle a second type of data.

**Input Format 1 - With a dataclass** This format corresponds to the case where snippets of input code are I/O based, such as in the following examples:

```
#A. Array
n = int(input())
a,b,c =[],[],[]
l = list(map(int,input().split()))
for i in l:
    if i<0:
        a.append(i)
    elif i>0:
        b.append(i)
    else:
        c.append(i)

if len(b)==0 and len(a)>2:
    b.append(a.pop())
    b.append(a.pop())
if len(a)%2==0:
    c.append(a.pop())
print(len(a),*a)
print(len(b),*b)
print(len(c),*c)
```

In which case the corresponding input example, as given to the complexity framework, will be formatted in the following manner:

```
'4\n-1 -2 -3 0\n'
```

This is the case where the input, whatever the number of distinct arguments there really is, is concatenated as a single string. In this case, the framework could not alone guess where the arguments are, without context, especially as it is not relying on any LLM. The framework, in order to run, needs an external dataclass that specifies how to understand the input string and cut it into different arguments, that the framework can then try to change to understand the time and space dependencies upon each of them.

Using a LLM, we can first infer the dataclass corresponding to a particular code challenge, before using it as part of the complexity framework in order to parse the input example and perform the various measurements on the variations of the inputs. Such a dataclass will have the following format:

```
@dataclass
class Input:
    n: int
    a_list: List[int]

    @classmethod
    def from_str(cls, input_: str):
        n, a_list, _ = input_.split('\n')
        n = int(n)
        a_list = list(map(int, a_list.split()))
        assert n == len(a_list)
        return cls(n, a_list)
```

```
def __repr__(self):
    return str(self.n) + '\n' + ' '.join(map(str, self.a_list)) + '\n
'
```

**Input Format 2 - Standalone code snippet**  This second format corresponds to the case where the input code is call-based, that is to say a particular function is being executed on a particular set of inputs, in which case the framework does have the information of how the inputs are separated into different arguments to the function (and that will also parametrize the final time and space complexity). Such call-based code generally has the following aspect:

```
class Array_300_A:
def solve(self, n, l):
    #A. Array
    a,b,c =[],[],[]
    for i in l:
        if i<0:
            a.append(i)
        elif i>0:
            b.append(i)
        else:
            c.append(i)

    if len(b)==0 and len(a)>2:
        b.append(a.pop())
        b.append(a.pop())
    if len(a)%2==0:
        c.append(a.pop())
    print(len(a),*a)
    print(len(b),*b)
    print(len(c),*c)
```

It is accompanied by inputs of the form:

```
{"n": "4", "l": "[-1, -2, -3, 0]"}
```

The framework can in this case directly understand the structure of the input example, and based on that infer the complexity of the code snippet.

### G.3  FUZZING AND WORST RUNNING CASES

BIGO(BENCH) focuses on worst-case time and space complexities, therefore any LLM-generated code solution is evaluated to infer its tightest upper-bound time and space complexities. To do so, any code solution input to the framework comes with a corresponding edge case input, as shared in the corresponding dataset that we made available. Using fuzzing, this edge case input is derived into many inputs of different sizes following different generators: for example, an input pair consisting of an integer and a list can see the integer input grow independently from the list (the integer can grow linearly, with random size steps, etc, whereas the list can remain static) or on the contrary interdependently (both the integer and the list grow at a regular pace, with same size steps). In the Github repository also made available, src/complexity/input_generations is the module in charge of handling the fuzzing. Then, the sandbox and the time and memory profilers record the behavior of the code solution on all sets of inputs, a set of inputs being parametrized by a specific input generator suggesting inputs over a range of sizes. This enables to gather time execution and memory footprint curves over many input cases, and worst execution curves are used to derive the form of the associated complexity using curve fitting methods, in the module src/complexity/curve_fitting.

Fuzzing is in itself a whole area of research, with many publications covering Java or C++ programs (Noller et al., 2018; Wang et al., 2019). More recently, LLM capabilities are more and more studied as a way of making fuzzing techniques more exhaustive (Xia et al., 2024; Xu et al., 2025). In our benchmark, we chose to design an evaluation framework completely independent from any machine learning model, so as to avoid biasing the evaluation. Nevertheless, the modularity of the codebase we release enables to switch the default fuzzer so to experiment with any other fuzzer released by the computer science community, or more recently from LLM-based approaches. This would

enable to measure what kind of bias can arise, and whether coverage and accuracy of the complexity evaluation are improved. Finally, without changing the fuzzer, the input edge cases example can also be interchanged easily as it is an input of the framework fuzzer: one can experiment with prompting a LLM for more edge case inputs.

Our paper provides methods to evaluate the quality of the fuzzing as performed by our evaluation framework, using ground-truth human solutions to all BigOBench benchmark problems. As detailed in Section 4.3, we conducted a thorough review to evaluate whether our evaluation framework correctly identifies worst-case scenarios and accurately derive complexity estimates from them, using a total of 250 samples, including 125 for time complexity and 125 for space. On the time complexity test set, the framework reaches 84% accuracy, with 1000-bootstrap samples confidence interval [0.776, 0.904] (for space, 82% accuracy [75.2, 88.8]). In addition, these test sets include, for time complexity, 42% problems of difficulty A (for space 45%), 29% of B (for space 25%) and 30% of C+ (for space 30%). On hard problems C+, framework accuracy is 84% for time complexity (84% for space). Being exposed to stochastic noise when measuring runtimes and memory footprints, the evaluation framework also got evaluated for its consistency, running the framework 20 times on 10 solutions of every problem and complexity class of the candidate test set: self-consistency of the framework is measured to be at 91.9% (resp. 89.1%) for time (resp. space) complexity, for a total of 10,130 (resp. 10,520) different code solutions.

Section G provide an illustration and more details concerning the methods being used in the framework in order to measure worst-case complexities.

### G.4 VARIANCE REDUCTION TECHNIQUES

Variance reduction techniques are important as the complexity framework relies on empirical measures of runtime and memory footprint, which are subject to noise. The framework was tested and ablated on a validation set in order to measure the effect of each design choice. Among them, each measure of runtime and memory footprint is repeated R times (in the current setting, R = 10 after ablation, trading-off accuracy with added compute cost) and then aggregated following a variance reduction technique. It turned out that the min-aggregation provided the best results for runtime measurements, improving complexity framework by 3.5% over median aggregation and 19% over max aggregation, among others. Our intuition is that runtimes are typically subject to variability caused by external factors (such as background processes, CPU throttling, system load etc.) in a way that increases runtime, creating high outliers.

The Hodges-Lehmann estimator (Hodges Jr. & Lehmann, 1963; Qiu et al., 2025) for example has not been specifically tested yet for the complexity framework. The codebase was designed with modularity as a core principle and any estimator can be added to src/complexity/curve_fitting/fitting_curve.py as a function of a set of empirical measures.

Beyond the variance-reduction techniques that can be used on empirical noisy measures, BIGO(BENCH) also employs similar technique when aggregating scores over multi-samples. Qiu et al. (2025) employs a Rao-Blackwellized bootstrap estimator, while we use an unbiased estimator for @k measures by leveraging n = 20 samples for k <= 10 with c <= n correct samples:

$$\text{pass@}k = \mathbb{E}_p \left[ 1 - \frac{\binom{n-c}{k}}{\binom{n}{k}} \right]$$

This estimator, introduced in Chen et al. (2021a), is an unbiased measure of the multi-sample performance as demonstrated in the appendix A of this paper.

In general, given that as measured in Section 4.3 we measure the accuracy of the framework with human annotations at ∼85%, we estimated that this success rate ensured the framework was accurate enough to provide reliable conclusions on the performance of LLMs as part of the BIGO(BENCH).

## G.5 COMPLEXITY FRAMEWORK PARAMETRIZATION

The complexity framework is parameterized by a suite of arguments that govern its execution, categorized into several domains. Dataset-related arguments specify the input data, including the required `path_to_jsonl_file` and optional `sub_key` for nested dictionaries, alongside indices (`code_start_index`, `code_end_index`) and filters (`filter_on_problem`, `multiply_samples_factor`) to control code selection and replication. Input handling is configured via `input_handler`, while logging and output behaviors (`log_outputs`, `save_results`, `skip_saving_full_results`, `results_folder_name_root`) dictate result persistence. Measurement parameters (`shuffle_runs`, `correct_nlogn`, `multiplier_op`, `multiplier_start`, `multiplier_repeat`, `multiplier_end`, `multiplier_mult_step`, `multiplier_max_increase`, `size_of_other_arguments`, `time_profiler`) regulate input scaling and timing. Complexity fitting is modulated by outlier removal (`filter_outliers`), penalty/constraint application (`apply_penalty`, `apply_constraints`), and aggregation strategies (`aggregate_y_values`, `max_time_rate`, `elect_complexity_time`, `elect_complexity_space`, `fix_constant_complexity`, `fix_negligeable_complexity`). Resource management includes memory/time thresholds (`temp_file_name_seed`, `memory_limit`, `timeout`, `large_timeout`, `giga_timeout`, `global_timeout`) and CPU allocation policies (`max_workers`, `number_solutions_per_worker`, `main_process_cpu_id_list`, `forkserver_type`, `use_distinct_forkservers`, `forkserver_cpu_id_list`). SLURM-specific arguments in `slurm.sh` further tailor high-performance computing deployment.

## G.6 IMPLEMENTATION LIMITATIONS AND FUTURE DIRECTIONS

The Dynamic Complexity Inference Framework can be improved in many ways. The current version that we release is more of a Proof-of-Concept, trying to see whether such framework can be used reliably on this task, for evaluation purposes of reinforcement learning. The following suggestions on how to improve the performance, reliability and maintainability of the framework are listed:

1. Refactoring the whole framework: the goal being to allow for flexible extensions. Some parts of the codebase, for instance around the input generations, are already designed to allow new methods of input generation.

2. Fuzzing: The modularity of the codebase presents opportunities for future research directions. One potential area of investigation is the exploration of alternative fuzzers, including those leveraging large language models (LLMs), to measure the introduction of bias and evaluate their impact on coverage and accuracy of complexity evaluation. Additionally, researchers could investigate the effectiveness of using LLMs to generate input edge cases, potentially leading to improved coverage and accuracy. Furthermore, the framework's modularity enables the study of the impact of different fuzzers on the codebase, allowing for a deeper understanding of the relationships between fuzzing strategies, input edge cases, and complexity evaluation outcomes. By experimenting with different fuzzers and input edge cases, researchers can gain insights into the strengths and limitations of various approaches, ultimately contributing to the development of more robust and effective fuzzing techniques.

3. Noise reduction: several methods can help with noise reduction, and therefore better accuracy of the framework as well as more stable results. Deterministic CPU operations is a huge axis of improvement, but post-processing methods on the runtime and memory footprint measures is also promising. Variance reduction techniques such as Hodges-Lehmann estimator could also be used.

4. Complexity definition: work on the definition of complexity and therefore how complexity is induced by the framework. Current assumptions may not be accurate, and maybe some choices in the framework implementation and not coherent with more widely spread definitions of complexity of a code snippet.

5. Complexity coverage: The way we designed the framework is that it defines a set of base functions in its module src/complexity/curve_fitting/fitting_class.py, and the framework can combine these functions with addition and multiplication operators to best fit the time execution and memory footprint curves.

   As long as such a function is defined, it is fit by the complexity framework. In practice, currently there is a class for the cubic function, and no classes that represent $x \to \frac{x^3}{\log x}$, $x \to (\log x)^6$ nor $x \to 2^{\text{polylog}(x)}$

   In practice, accuracy limitations may arise when two classes exhibit similar behavior over typical input sizes (1-10,000). Empirical noise from profiler measurements (runtime or memory footprint) can make fitting complexities challenging. We recommend testing the framework on labeled examples when introducing a new function class.

6. Pure performance: the performance of the complexity framework is limited. Measurement of these limitations and their improvement is the priority goal.

## H    BENCHMARK EXPERIMENTAL SETUP

The experiments conducted in this paper were done on an internal cluster of H100 GPUs. Running the BIGO(BENCH) evaluation on low-compute models that do not produce reasoning tokens, e.g. the LLAMA 4 models, with @10 metrics for the three tasks of complexity prediction, generation and ranking, both on time and space complexity test sets, required 100 GPU hours. On the contrary, more compute-intensive workflows involving reasoning models, e.g. QWEN QWQ, necessitates up to 2500 GPU hours.

Using the solutions generated by a LLM, the BIGO(BENCH) evaluation framework can be run on a set of $n$ CPUs in order to perform the runtime and memory footprint measures, based on which a time-space complexity estimate is attributed to the code solution. For each LLM being benchmarked on BIGO(BENCH), hundreds of CPU hours can be required in total to run the framework on the time/space generation/ranking tasks, though this can highly vary depending on its parametrization: range of the measures, replication rate of each measure, ... In our most compute intensive setting (up to $3000\times$ the original input size and 10 measure replicas), we used up to 10k CPU hours.

Finally, as part of our fine-tuning experiments, we used for each of the four fine-tunings of LLAMA 3.1 70B, that is to say on time prediction, space prediction, time generation and space generation data, 120 GPU hours. These models were then evaluated based on the low-compute set-up, and finally the generated solutions were evaluated using compute for the complexity framework as detailed above.

Preliminary experiments for the complexity framework, that only require CPUs, used more compute than for the final runs used to evaluate the models on BIGO(BENCH). This is because as detailed in Section 3 ablations were conducted to find the optimal set of parameters for the framework, given the resources we had access to; we insist on the fact that the framework can be run with much lower compute by adjusting various parameters such as its range of measures or its replication rate. On the contrary, for the GPU compute, which comes from running the inference on a variety of models being benchmarked by BIGO(BENCH), the majority of it corresponds directly to the experiments being reported in this paper.

In total, the experiments reported in this paper used 12,000 GPU hours for model fine-tuning and inference. In order to run the evaluation framework on all tasks of all models, the experiments used 180,000 CPU hours, though this number can be easily divided by a factor of 100 down to 1,800 CPU hours when using a less compute-intensive framework parametrization.

Concerning the results of these experiments being reported in the paper, we precise that only DEEPSEEK-R1 distilled models are reported, and that DEEPSEEK-R1 is not reported as an initial assessment led to over-budget compute usage; this partial run gave similar results on complexity tasks as DEEPSEEK-R1 LLAMA 70B.

The table results that correspond to the BIGO(BENCH) benchmarking of models are supported by one-tailed paired t-tests on 1000 bootstraps samples of the model results evaluate the significance of the superiority of the best model. The application of one-tailed paired t-tests on bootstrap samples assumes that the differences in model performance metrics (e.g., pass@1 scores) across the benchmark tasks are approximately normally distributed, an assumption that is bolstered by the Central Limit Theorem given the large number of bootstrap iterations (1000). Fig. 11 provides an overview of the distribution of differences in model performance metrics. Additionally, the tests presume that the paired nature of the data—where each model is evaluated on the same set of tasks—is preserved through resampling, ensuring that dependencies between model outputs are accounted for.

Any @k metric uses an unbiased estimator based on 20 samples.

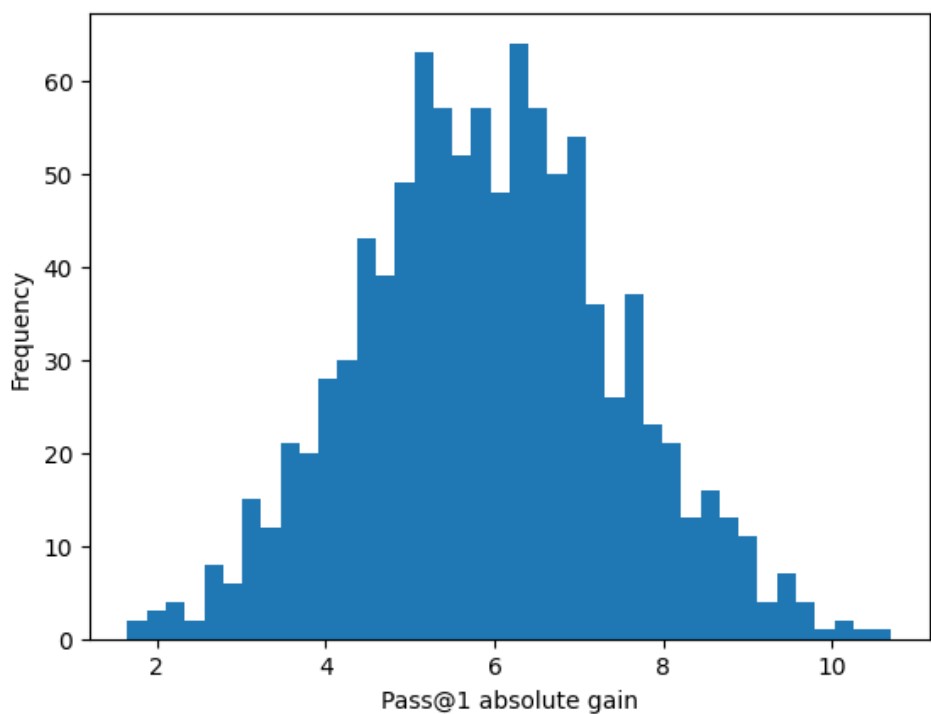

Figure 11: $Pass@1$ absolute gain of DEEPSEEK R1 LLAMA 70B over LLAMA 3.3 70B for the task of time complexity prediction, over 1000 bootstrap samples of BIGO(BENCH) time complexity test set.

# I  LICENSES

| Code Repository/Dataset | License |
|---|---|
| CodeContests | Apache 2.0 [link] |
| big_O | BSD-3 [link] |
| llama 4 models | Llama 4 License |
| llama 3.3 | Llama 3.3 License |
| Gemma 3 27B | Gemma License |
| Codestral 22B | MNPL 0.1 License |
| Qwen2.5-Coder 32B | Apache 2.0 [link] |
| DeepseekCoderV2 236B | Deepseek License |
| DeepseekV3 671B | Deepseek License Agreement |
| DeepseekR1 Qwen 32B | MIT License [link] |
| DeepseekR1 Llama 70B | MIT License [link] |
| Llama 3.1 Nemotron-Ultra 253B | NVIDIA Open License Agreement |
| Qwen QwQ 32B | Apache 2.0 [link] |
| Qwen3 32B | Apache 2.0 [link] |

Table 13: Licenses of Code Repositories and Datasets

Table 13 lists all licenses of code and data that were used in this paper, along with licenses of models that were downloaded and run locally to be benchmarked for this paper.

# J  COMPLEXITY PREDICTION EXAMPLE

## J.1  EXAMPLE OF QUERY

Provide the time complexity for the following competitive programming question and corresponding solution. When analyzing the complexity of an algorithm, consider the worst-case scenario where all possible input combinations are tried, given the following conditions: 1. the inputs must adhere to the specified data types of the problem; 2. the inputs should not cause the code to crash or exit on an exception; 3. the inputs do not necessarily need to satisfy additional constraints that are potentially mentioned in the problem statement; 4. calling input() does not consume runtime nor memory, but of course any operations on top of it or afterwards will be counted towards runtime and memory footprint; 5. Anything printed gets added to the memory. You can take advantage of Python-specific optimizations provided by the underlying CPython interpreter or compiler to achieve the desired complexity, and you must account for them when analyzing the complexity. Here is the programming question:

You are given n segments on a number line; each endpoint of every segment has integer coordinates. Some segments can degenerate to points. Segments can intersect with each other, be nested in each other or even coincide.

The intersection of a sequence of segments is such a maximal set of points (not necesserily having integer coordinates) that each point lies within every segment from the sequence. If the resulting set isn't empty, then it always forms some continuous segment. The length of the intersection is the length of the resulting segment or 0 in case the intersection is an empty set.

For example, the intersection of segments [1;5] and [3;10] is [3;5] (length 2), the intersection of segments [1;5] and [5;7] is [5;5] (length 0) and the intersection of segments [1;5] and [6;6] is an empty set (length 0).

Your task is to remove exactly one segment from the given sequence in such a way that the intersection of the remaining (n - 1) segments has the maximal possible length.

Input

The first line contains a single integer $n$ $(2 \leq n \leq 3 \times 10^5)$  the number of segments in the sequence.

Each of the next n lines contains two integers $l_i$ and $r_i$ $(0 \leq l_i \leq r_i \leq 10^9)$  the description of the ith segment.

Output

Print a single integer - the maximal possible length of the intersection of (n - 1) remaining segments after you remove exactly one segment from the sequence.

Examples

Input

4 1 3 2 6 0 4 3 3

Output

1

Input

5 2 6 1 3 0 4 1 20 0 4

Output

2

Input

3 4 5 1 2 9 20

Output

0

Input

2 3 10 1 5

Output

7

Note

In the first example you should remove the segment [3;3], the intersection will become [2;3] (length 1). Removing any other segment will result in the intersection [3;3] (length 0).

In the second example you should remove the segment [1;3] or segment [2;6], the intersection will become [2;4] (length 2) or [1;3] (length 2), respectively. Removing any other segment will result in the intersection [2;3] (length 1).

In the third example the intersection will become an empty set no matter the segment you remove.

In the fourth example you will get the intersection [3;10] (length 7) if you remove the segment [1;5] or the intersection [1;5] (length 4) if you remove the segment [3;10].

Here is the corresponding Python solution:

```python
n = int(input())
L = []
R = []
S = []
for _ in range(n):
    a,b = [int(x) for x in input().split()]
    L.append(a)
    R.append(b)
    S.append((a,b))

L.sort(reverse = True)
R.sort()

if (L[0],R[0]) in S:
    print(max(R[1]-L[1],0))
else:
    print(max(R[0]-L[1],R[1]-L[0],0)).
```

Please ignore any constraints on the input sizes that may have been previously mentioned in the problem description. Compute the big-O complexity as if inputs can be as large as possible. Output the big-O time complexity only, no explanation needed, no other words needed.

## J.2    EXAMPLE OF RESPONSE FROM DEEPSEEK-R1 LLAMA 70B

*<think>*

Okay, I need to figure out the time complexity of the given Python solution for the problem. Let's go through the code step by step.

First, the code reads n, which is the number of segments. Then, for each of the next n lines, it reads two integers l_i and r_i, appending them to lists L and R, and storing the tuples in S.

So the initial part is O(n) time because each of the n lines is processed once.

Next, the code sorts L in descending order. Sorting a list of size n in Python using the built-in sort function is O(n log n). Similarly, R is sorted in ascending order, which is also O(n log n).

After sorting, there's a condition checking whether the tuple (L[0], R[0]) is in S. Checking membership in a list in Python is O(n) because it has to potentially scan all elements. So this check is O(n).

Then, depending on the condition, it either prints max(R[1] - L[1], 0) or max(R[0] - L[1], R[1] - L[0], 0). Both of these operations are O(1) since they involve a couple of subtractions and a max function.

So putting it all together:

- Reading the inputs: O(n) - Sorting L and R: O(n log n) each, so total O(2n log n) which simplifies to O(n log n) - Checking the tuple in S: O(n) - The rest is O(1)

The dominant term here is O(n log n) from the sorting steps. The other steps are either O(n) or O(1), which are lower order terms.

Therefore, the overall time complexity is O(n log n).

$<\backslash think>$

The time complexity is O(n log n).

O(n log n)

# K  COMPLEXITY GENERATION EXAMPLE

## K.1  EXAMPLE OF QUERY

Provide a Python solution for the following competitive programming question: You are given n segments on a number line; each endpoint of every segment has integer coordinates. Some segments can degenerate to points. Segments can intersect with each other, be nested in each other or even coincide.

The intersection of a sequence of segments is such a maximal set of points (not necesserily having integer coordinates) that each point lies within every segment from the sequence. If the resulting set isn't empty, then it always forms some continuous segment. The length of the intersection is the length of the resulting segment or 0 in case the intersection is an empty set.

For example, the intersection of segments [1;5] and [3;10] is [3;5] (length 2), the intersection of segments [1;5] and [5;7] is [5;5] (length 0) and the intersection of segments [1;5] and [6;6] is an empty set (length 0).

Your task is to remove exactly one segment from the given sequence in such a way that the intersection of the remaining (n - 1) segments has the maximal possible length.

Input

The first line contains a single integer $n$ $(2 \leq n \leq 3 \times 10^5)$ the number of segments in the sequence.

Each of the next n lines contains two integers $l_i$ and $r_i$ $(0 \leq l_i \leq r_i \leq 10^9)$ the description of the ith segment.

Output

Print a single integer - the maximal possible length of the intersection of (n - 1) remaining segments after you remove exactly one segment from the sequence.

Examples

Input

4 1 3 2 6 0 4 3 3

Output

1

Input

5 2 6 1 3 0 4 1 20 0 4

Output

2

Input

3 4 5 1 2 9 20

Output

0

Input

2 3 10 1 5

Output

7

Note

In the first example you should remove the segment [3;3], the intersection will become [2;3] (length 1). Removing any other segment will result in the intersection [3;3] (length 0).

In the second example you should remove the segment [1;3] or segment [2;6], the intersection will become [2;4] (length 2) or [1;3] (length 2), respectively. Removing any other segment will result in the intersection [2;3] (length 1).

In the third example the intersection will become an empty set no matter the segment you remove.

In the fourth example you will get the intersection [3;10] (length 7) if you remove the segment [1;5] or the intersection [1;5] (length 4) if you remove the segment [3;10].

Output the code only. Generate code that has an algorithmic time complexity of $O(n.\log(n))$.

When analyzing the complexity of an algorithm, consider the worst-case scenario where all possible input combinations are tried, given the following conditions: 1. the inputs must adhere to the specified data types of the problem; 2. the inputs should not cause the code to crash or exit on an exception; 3. the inputs do not necessarily need to satisfy additional constraints that are potentially mentioned in the problem statement; 4. calling input() does not consume runtime nor memory, but of course any operations on top of it or afterwards will be counted towards runtime and memory footprint; 5. Anything printed gets added to the memory. You can take advantage of Python-specific optimizations provided by the underlying CPython interpreter or compiler to achieve the desired complexity, and you must account for them when analyzing the complexity.

Your code should be enclosed in triple backticks like so: "'python YOUR CODE HERE "'. Use the backticks for your code only.

### K.2 Example of response from DeepSeek-R1 Llama 70B

*<think>*

Okay, I've got this programming problem to solve. Let me read it carefully.

So the problem is about segments on a number line. Each segment has integer endpoints, and some might be just points. We need to remove exactly one segment such that the intersection of the remaining n-1 segments is as long as possible. The output is the maximum possible length of this intersection.

Hmm, the intersection of multiple segments is the maximum set of points that lie in all of them. So for each segment, if I remove it, I need to compute the intersection of the others and find which removal gives the maximum length.

The challenge is to do this efficiently, especially since n can be up to 3e5, which is 300,000. So any O(n^2) approach is out of the question. I need an O(n log n) solution.

Let me think about what determines the intersection of all segments except one.

The intersection of all segments except one is determined by the maximum of the left endpoints and the minimum of the right endpoints. So, for each i, if I remove the i-th segment, the new intersection would be [max_left, min_right], where max_left is the maximum of all l's except l_i, and min_right is the minimum of all r's except r_i. The length is max(0, min_right - max_left).

So the problem reduces to, for each i, compute the maximum of l's excluding l_i, and the minimum of r's excluding r_i, then calculate the difference.

But how do I compute these efficiently?

One approach is to precompute prefix and suffix maximums for the left endpoints and prefix and suffix minimums for the right endpoints.

Wait, let me think:

For each position i, the max left when excluding i can be found by taking the maximum of the prefix up to i-1 and the suffix starting from i+1. Similarly for the min right.

Yes, that makes sense.

So, let's outline the steps:

1. Read all the segments into two lists: lefts and rights.

2. Precompute prefix_max_left: an array where prefix_max_left[i] is the maximum of lefts[0..i].

3. Precompute suffix_max_left: an array where suffix_max_left[i] is the maximum of lefts[i..n-1].

4. Similarly, compute prefix_min_right and suffix_min_right.

Wait, no. Wait, for prefix_max_left, it's the maximum up to i, but for the suffix, it's the maximum from i to the end. But when we want to exclude the i-th element, the max left is the maximum of all elements except the i-th. So, for each i, the max left is max( prefix_max_left[i-1], suffix_max_left[i+1] ). Similarly for the min right.

So, to compute the max left when excluding i, we take the maximum of the prefix up to i-1 and the suffix starting at i+1.

Similarly, the min right when excluding i is the min of the prefix up to i-1's min and the suffix starting at i+1's min.

So, let's think about how to precompute these prefix and suffix arrays.

For the prefix_max_left:

prefix_max_left[0] = lefts[0] prefix_max_left[i] = max(prefix_max_left[i-1], lefts[i])

Similarly, suffix_max_left[n-1] = lefts[n-1] suffix_max_left[i] = max(suffix_max_left[i+1], lefts[i])

Same logic applies to prefix_min_right and suffix_min_right, but using min instead.

Wait, no. For prefix_min_right:

prefix_min_right[0] = rights[0] prefix_min_right[i] = min(prefix_min_right[i-1], rights[i])

And for suffix_min_right:

suffix_min_right[n-1] = rights[n-1] suffix_min_right[i] = min(suffix_min_right[i+1], rights[i])

Once I have these arrays, for each i, the max left excluding i is:

if i is 0: suffix_max_left[1] if i is n-1: prefix_max_left[n-2] else: max(prefix_max_left[i-1], suffix_max_left[i+1])

Similarly for the min right.

Then, for each i, calculate the possible length as max(0, min_right - max_left). Keep track of the maximum such value.

That seems manageable.

Now, let's think about the steps in code:

- Read n. - Read n lines, collect lefts and rights. - Compute prefix_max_left and suffix_max_left. - Compute prefix_min_right and suffix_min_right. - For each i, compute max_left and min_right as described. - Calculate the length, and track the maximum.

But wait, what about when n=2? For example, when i is 0, the suffix_max_left is suffix_max_left[1], which is lefts[1]. Similarly, when i is 1, the prefix_max_left is prefix_max_left[0], which is lefts[0]. So that case is handled correctly.

Let's test the approach against the sample inputs.

First sample:

Input: 4 1 3 2 6 0 4 3 3

So lefts = [1,2,0,3] rights = [3,6,4,3]

Compute prefix_max_left: prefix[0] = 1 prefix[1] = max(1,2)=2 prefix[2] = max(2,0)=2 prefix[3] = max(2,3)=3

suffix_max_left: suffix[3] =3 suffix[2] = max(0,3)=3 suffix[1] = max(2,3)=3 suffix[0] = max(1,3)=3

prefix_min_right: prefix[0] =3 prefix[1] = min(3,6)=3 prefix[2] = min(3,4)=3 prefix[3] = min(3,3)=3

suffix_min_right: suffix[3]=3 suffix[2] = min(4,3)=3 suffix[1] = min(6,3)=3 suffix[0] = min(3,3)=3

Now, for each i:

i=0: max_left = suffix_max_left[1] =3 min_right = suffix_min_right[1] =3 length is 0.

i=1: max_left = max(prefix_max_left[0], suffix_max_left[2])=max(1,3)=3 min_right = min(prefix_min_right[0], suffix_min_right[2])= min(3,3)=3 length 0.

i=2: max_left = max(prefix_max_left[1] (2), suffix_max_left[3] (3)) =3 min_right = min(prefix_min_right[1] (3), suffix_min_right[3] (3)) =3 length 0.

i=3: max_left = prefix_max_left[2] =2 min_right = prefix_min_right[2] =3 so 3-2=1.

So the maximum is 1, which matches the sample.

Another sample:

Input: 2 3 10 1 5

lefts = [3,1] rights = [10,5]

prefix_max_left: [3,3] suffix_max_left: [3,1]

prefix_min_right: [10,5] suffix_min_right: [5,5]

For i=0: max_left is suffix_max_left[1] =1 min_right is suffix_min_right[1] =5 length 5-1=4.

For i=1: max_left is prefix_max_left[0]=3 min_right is prefix_min_right[0]=10 length 7.

So maximum is 7, which is correct.

So the approach seems correct.

Now, the code:

We can implement this in Python.

We'll read n, then the list of segments.

Compute prefix_max_left, suffix_max_left, prefix_min_right, suffix_min_right.

Then loop through each i, compute the max left and min right when excluding i.

Compute the length and keep the maximum.

Now, let's think about handling the cases where i is at the beginning or end.

For example, when i=0, the suffix_max_left[i+1] is suffix_max_left[1], and the prefix_max_left[i-1] is nothing (since i-1 is -1). So for i=0, the max left is suffix_max_left[1], and the min right is suffix_min_right[1].

Similarly, for i =n-1, the suffix is nothing, so the max left is prefix_max_left[n-2], and the min right is prefix_min_right[n-2].

So in code, for each i:

if i ==0: current_max_left = suffix_max_left[1] current_min_right = suffix_min_right[1] elif i ==n-1: current_max_left = prefix_max_left[n-2] current_min_right = prefix_min_right[n-2] else: current_max_left = max(prefix_max_left[i-1], suffix_max_left[i+1]) current_min_right = min(prefix_min_right[i-1], suffix_min_right[i+1])

Then compute current_length = max(0, current_min_right - current_max_left)

We can keep track of the maximum.

Now, let's think about the implementation.

In Python:

n = int(input()) segments = [] lefts = [] rights = [] for _ in range(n): l, r = map(int, input().split()) lefts.append(l) rights.append(r)

Compute prefix_max_left:

prefix_max = [0]*n prefix_max[0] = lefts[0] for i in range(1, n): prefix_max[i] = max(prefix_max[i-1], lefts[i])

Compute suffix_max:

suffix_max = [0]*n suffix_max[-1] = lefts[-1] for i in range(n-2, -1, -1): suffix_max[i] = max(lefts[i], suffix_max[i+1])

Same for prefix_min_right:

prefix_min = [0]*n prefix_min[0] = rights[0] for i in range(1, n): prefix_min[i] = min(prefix_min[i-1], rights[i])

suffix_min:

suffix_min = [0]*n suffix_min[-1] = rights[-1] for i in range(n-2, -1, -1): suffix_min[i] = min(rights[i], suffix_min[i+1])

Then, for each i in 0..n-1:

if i ==0: ml = suffix_max[1] if n¿1 else 0 mr = suffix_min[1] if n¿1 else 0 elif i ==n-1: ml = prefix_max[n-2] if n¿1 else 0 mr = prefix_min[n-2] if n¿1 else 0 else: ml = max(prefix_max[i-1], suffix_max[i+1]) mr = min(prefix_min[i-1], suffix_min[i+1])

Wait, but what if n=1? Oh, but according to the problem statement, n is at least 2. So n is ¿=2, so no problem.

So code:

max_length = 0 for i in range(n): if i ==0: ml = suffix_max[1] mr = suffix_min[1] elif i == n-1: ml = prefix_max[n-2] mr = prefix_min[n-2] else: ml = max(prefix_max[i-1], suffix_max[i+1]) mr = min(prefix_min[i-1], suffix_min[i+1]) current_length = max(0, mr - ml) if current_length ¿ max_length: max_length = current_length

print(max_length)

Wait, but wait: in the case where all segments except one have a higher max left than the min right, the current_length could be negative, but we take max with 0.

Yes.

Testing this logic with the first sample:

n=4, i=3 (last element). So i=3 is n-1.

ml is prefix_max[2] =2.

mr is prefix_min[2] =3.

3-2=1. So correct.

Another test case: when n=2.

i=0: ml is suffix_max[1] =1, mr is suffix_min[1] =5.

5-1=4.

i=1: ml is prefix_max[0]=3, mr is prefix_min[0]=10.

10-3=7.

So maximum is 7.

Another test case:

Input:

3 4 5 1 2 9 20

lefts = [4,1,9] rights = [5,2,20]

prefix_max:

[4,4,9]

suffix_max:

[4,9,9]

prefix_min:

[5,2,2]

suffix_min:

[2,2,20]

Now, for each i:

i=0: ml = suffix_max[1] =9, mr = suffix_min[1] =2. 2-9 is -7, so 0.

i=1: ml is max(prefix_max[0] (4), suffix_max[2] (9)) =9.

mr is min(prefix_min[0] (5), suffix_min[2] (20)) =5.

5-9 is -4, 0.

i=2: ml is prefix_max[1] =4.

mr is prefix_min[1] =2.

2-4 is -2, 0.

So maximum is 0.

Which matches the sample.

So the code seems correct.

Now, considering the time complexity.

Each of the prefix and suffix arrays are computed in O(n) time. Then, for each i, it's O(1) operations. So overall O(n) time, which is acceptable for n up to 3e5.

So this should work.

Now, let's code this.

Wait, but in Python, for n=2, the suffix_max for i=1 is lefts[1], which is correct.

Yes.

Now, code:

Read n.

Read n lines, collect lefts and rights.

Compute prefix_max, suffix_max, prefix_min, suffix_min.

Loop through each i, compute ml and mr.

Compute current_length.

Update max_length.

Finally, print max_length.

So, the code is as follows.

$<\backslash think>$

To solve this problem, we need to determine the maximum possible length of the intersection of segments after removing exactly one segment from the given sequence. The approach involves efficiently computing the intersection of the remaining segments for each possible removal and selecting the maximum length.

1. **Reading Input:** Read the number of segments and their endpoints. 2. **Prefix and Suffix Arrays:** Compute prefix and suffix arrays for both left and right endpoints: - 'prefix_max_left' and 'suffix_max_left' to track the maximum left endpoints up to and from each position. - 'prefix_min_right' and 'suffix_min_right' to track the minimum right endpoints up to and from each position. 3. **Compute Intersection for Each Removal:** For each segment, compute the intersection length when that segment is removed using the precomputed prefix and suffix arrays. The intersection length is determined by the maximum left endpoint and minimum right endpoint of the remaining segments. 4. **Track Maximum Length:** Keep track of the maximum intersection length encountered.

'''python

```python
n = int(input())
lefts = []
rights = []
for _ in range(n):
    l, r = map(int, input().split())
    lefts.append(l)
    rights.append(r)

prefix_max = [0] * n
prefix_max[0] = lefts[0]
for i in range(1, n):
    prefix_max[i] = max(prefix_max[i-1], lefts[i])

suffix_max = [0] * n
suffix_max[-1] = lefts[-1]
for i in range(n-2, -1, -1):
    suffix_max[i] = max(lefts[i], suffix_max[i+1])

prefix_min = [0] * n
prefix_min[0] = rights[0]
for i in range(1, n):
    prefix_min[i] = min(prefix_min[i-1], rights[i])

suffix_min = [0] * n
suffix_min[-1] = rights[-1]
for i in range(n-2, -1, -1):
    suffix_min[i] = min(rights[i], suffix_min[i+1])

max_length = 0
for i in range(n):
    if i == 0:
        current_max_left = suffix_max[1]
        current_min_right = suffix_min[1]
    elif i == n-1:
        current_max_left = prefix_max[n-2]
        current_min_right = prefix_min[n-2]
    else:
        current_max_left = max(prefix_max[i-1], suffix_max[i+1])
        current_min_right = min(prefix_min[i-1], suffix_min[i+1])
    current_length = max(0, current_min_right - current_max_left)
    if current_length > max_length:
        max_length = current_length

print(max_length)
```

'''

1. **Reading Input:** The input is read and stored in two lists, 'lefts' for the left endpoints and 'rights' for the right endpoints. 2. **Prefix and Suffix Arrays:** These arrays help efficiently compute the maximum left and minimum right endpoints when excluding each segment: - 'prefix_max' and 'suffix_max' store the maximum left endpoints up to each index and from each index onwards,

respectively. - 'prefix_min' and 'suffix_min' store the minimum right endpoints up to each index and from each index onwards, respectively. 3. **Compute Intersection:** For each segment, compute the intersection of the remaining segments by excluding the current segment. This is done using the precomputed prefix and suffix arrays to find the maximum left and minimum right endpoints efficiently. 4. **Track Maximum Length:** The maximum length of the intersection is updated each time a valid intersection is found, ensuring the result is the longest possible intersection after removing exactly one segment.

This approach ensures an efficient solution with a time complexity of O(n), making it suitable for large input sizes up to 300,000 segments.

### K.3    NOTE ON THE LLM SOLUTION

Using public and private tests, the solution given by the LLM was evaluated as correct. Using the complexity framework, it was determined it is running in O(n) time, which does not fulfill the $O(n.\log(n))$ requirement.

## L REPARTITION OF COMPLEXITY CLASSES PER ALGORITHMIC NOTIONS

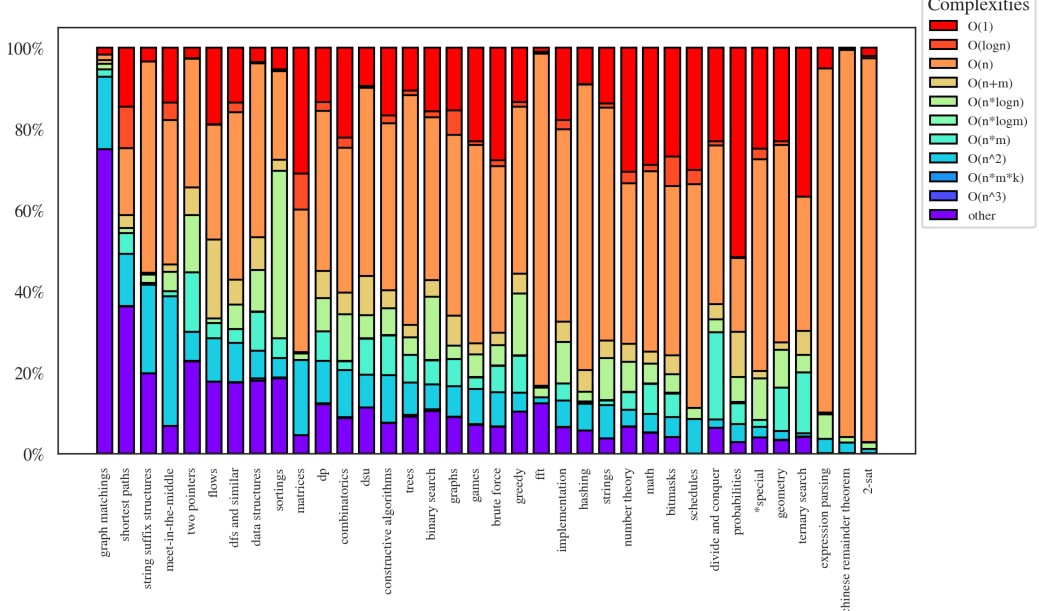

Figure 12: Breakdown of time complexity classes across different algorithmic categories. The stacked bar charts reveal how complexity requirements vary by problem type. Problems involving graph handling and string manipulation tend to have higher computational complexity, while basic arithmetic and sequence operations typically achieve more efficient complexity classes.

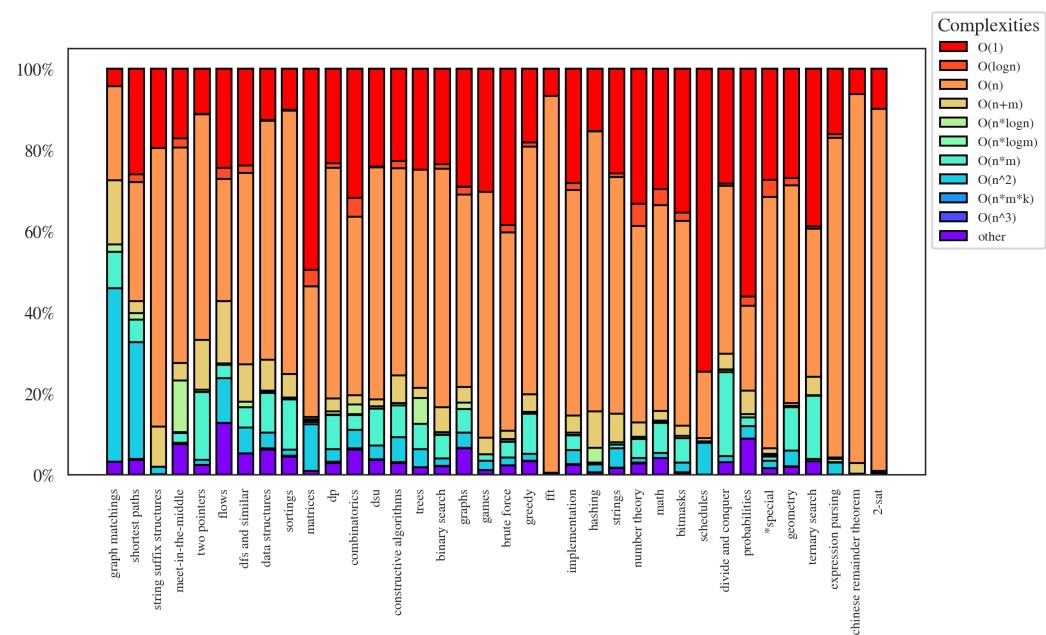

Figure 13: Breakdown of space complexity classes across different algorithmic categories. The stacked bar charts reveal how complexity requirements vary by problem type. Problems involving graph handling and string manipulation tend to have higher computational complexity, while basic arithmetic and sequence operations typically achieve more efficient complexity classes.

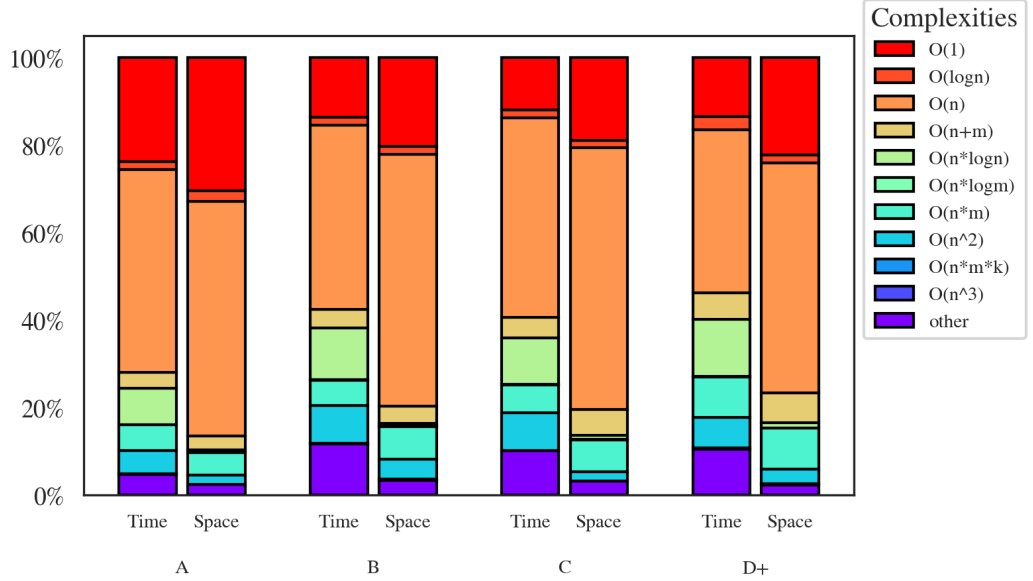

Figure 14: Evolution of time and space complexity distributions across problem difficulty levels (A through D+). This visualization demonstrates how harder problems tend to require more computationally intensive solutions. The proportion of linear and constant-time solutions decreases with difficulty, while the share of higher-order polynomial and logarithmic complexities increases.

## M    LLM USAGE

We employed LLMs as general-purpose writing assistance tools throughout the preparation of this paper. Specifically, LLMs were used to: (1) proofread text and correct grammatical errors, (2) refine and rewrite sentences and paragraphs for clarity and flow, (3) convert mathematical expressions and formulas into proper LaTeX format, (4) describe and explain implementation details of various components in our codebase, and (5) provide general assistance with academic writing structure and style. While LLMs contributed to the presentation and clarity of our work, all research ideas, methodologies, experimental design, and scientific contributions remain entirely the product of the listed authors.

