# OpenReview forum: "BigO(Bench) - Can LLMs Generate Code with Controlled Time and Space Complexity?"
_ICLR.cc/2026/Conference — Submitted to ICLR 2026_

### Official Review · Reviewer_vpRc · 2025-10-28

**Soundness:** 3
**Presentation:** 3
**Contribution:** 2
**Rating:** 2
**Confidence:** 3

**Summary:**

This paper introduces BIGO(BENCH), a benchmark designed to evaluate Large Language Models (LLMs) on their ability to understand and generate Python code with specific time and space complexity constraints. The contributions include: (1) A dataset derived from CODE CONTESTS, annotated with complexity labels using a custom dynamic inference framework (3,105 problems, ~1.2M solutions). (2) The complexity inference framework itself, which uses profiling, fuzzing, and curve fitting to estimate complexities. (3) An evaluation of 14 state-of-the-art LLMs on three tasks: complexity prediction, complexity-constrained code generation, and ranking generated solutions against human ones based on complexity coefficients. The results show that current LLMs, even reasoning-focused ones, struggle significantly with complexity generation tasks, despite high performance on standard code synthesis.

**Strengths:**

1. The paper addresses a critical and often overlooked aspect of code generation by LLMs – their understanding and control of computational complexity, which is crucial for real-world software development.

2. The authors have annotated a large dataset (3,105 problems, ~1.2M solutions) from competitive programming platforms with inferred complexity labels, providing a substantial resource if the labels were reliable.

3. The evaluation covers multiple facets of complexity handling (prediction, generation, ranking) and includes a wide range of recent LLMs.

4. The release of the complexity inference framework code allows for reproducibility and potential community improvement.

**Weaknesses:**

1. The reported 82-84% accuracy against human labels is insufficient for generating trustworthy ground truth. Reliance on empirical profiling makes the framework susceptible to noise, hardware variations, and specific runtime environments, potentially failing to capture true asymptotic complexity. The paper itself notes it can fall upon edge cases. This lack of robustness undermines the entire benchmark's validity.

2. The extensive, framework-dependent filtering applied during dataset creation may introduce significant bias, potentially selecting only problems/solutions where the framework performs well. The resulting dataset's representativeness is questionable. The complexity distribution is highly imbalanced, potentially skewing results.

3. The All@1 scores for complexity generation are exceptionally low (often <5% for time, <3% for space) across all models, including powerful reasoning models. While interpreted as LLM failure, these near-zero scores could equally indicate issues with the benchmark itself: unreliable ground truth, an overly harsh metric (All@k), or ill-posed tasks. It's hard to draw meaningful conclusions or measure progress when performance is near the floor.

4. Fine-tuning Llama 3.1 70B specifically on the benchmark tasks yielded negligible or even negative impacts on complexity generation performance (Table 4). This suggests the benchmark data and tasks, as currently formulated, may not provide a useful signal for improving LLM capabilities in this area, questioning their utility.

5. The framework relies on dynamic analysis (profiling runs). It's unclear how well this empirical approach approximates theoretical worst-case complexity, especially given Python's dynamic nature and CPython optimizations. The benchmark might be evaluating the ability to generate code that performs well empirically under the framework's specific testing conditions, rather than code with a provably correct asymptotic complexity.

6. Decisions like using Big-O for worst-case, the specific parameters for fuzzing/curve fitting, and the `simplicity bias`  need stronger justification and sensitivity analysis.

**Questions:**

1. Could the authors provide a more detailed error analysis for the complexity framework? What types of code or complexity classes does it struggle with most? How was the 125-sample human validation set selected, and what was the inter-annotator agreement if multiple humans were involved? Given the 16-18% error rate, how confident can we be in the benchmark labels, especially for evaluating subtle differences between models or measuring fine-tuning progress?

2. How sensitive are the framework's complexity estimations to the specific hardware, Python version, background processes, and profiling tool versions? Were experiments run to quantify this variability? Could this noise explain some of the poor LLM generation results?

3. Can the framework distinguish between empirical performance fitting a curve (e.g., $O(n)$ up to $n=10000$) and true theoretical complexity (e.g., an underlying $O(n \log n)$ algorithm that looks linear in the tested range)? How does it handle amortized analysis?

4. Could the extensive filtering applied during test set creation (e.g., removing outliers, unstable predictions, unlikely ASTs) have biased the benchmark towards problems/solutions the framework can easily analyze, potentially masking harder cases?

5. Given the near-zero All@k generation scores, how can the authors be sure this reflects inherent LLM limitations versus issues with the benchmark's noisy labels, task formulation, or the stringency of the All@k metric?

6. Why do the authors believe fine-tuning failed to improve (and sometimes worsened) complexity generation performance? Does this suggest the benchmark data lacks a useful signal, or that standard fine-tuning is simply inadequate for this type of reasoning?

7. Why was Big-O chosen for the main benchmark despite the potential ambiguity noted, especially when the Big-Theta prompt seemed to improve generation for the reasoning model?

---

> ### Author Response · Authors · 2025-11-21
>
> We thank reviewer vpRc for taking the time to provide a detailed review of our submission. In particular, we are thankful for the highlight on the importance of the task and the data/code release part of this benchmark.
>
> > “The reported 82-84% accuracy against human labels is insufficient for generating trustworthy ground truth. Reliance on empirical profiling makes the framework susceptible to noise, hardware variations, and specific runtime environments, potentially failing to capture true asymptotic complexity. The paper itself notes it can fall upon edge cases. This lack of robustness undermines the entire benchmark's validity.”
>
> We do not agree with the reviewer that the entire benchmark validity is undermined through these robustness and stability test results. As underlined in the paper, self-consistency was evaluated across 20 runs on ~10k solutions and led to 91.9% (time) and 89.1% (space) self-consistency. In terms of pure correctness, a human labeling effort to check the correctness of the test set labels reports 84% (time) and 82% (space) accuracy vs. human labels with confidence intervals [77.6%, 90.4%] and [75.2%, 88.8%]. In particular, the framework maintains accuracy of the labels even on hard problems. In general when performing evaluation runs with the framework, R=10 repetitions per measurement is performed to help reduce any potential noise in these runtime measurements. We do agree that the framework can be improved and this is an axis for future works, and we make suggestions such as using virtual CPU cores.
>
> In general, no such complexity data at scale was found to be available, especially across time and space complexity, across sufficient diversity of classes and coding problems in the first place (as discussed in the literature review).
>
> On top of this, even in this setting of synthetic labels, for which we explicitly study potential limitations and areas of improvements, top models fail short of reaching any significant scores, where any label noisiness could be considered as a limitation of their performance (top models being capped at 6.5 all@1 on time complexity generation for example). Even with imperfect labels, the benchmark remains discriminative and the empirical profiling helps in fact capture real-world performance.
>
> > “The extensive, framework-dependent filtering applied during dataset creation may introduce significant bias, potentially selecting only problems/solutions where the framework performs well. The resulting dataset's representativeness is questionable. The complexity distribution is highly imbalanced, potentially skewing results.”
>
> The filtering, as mentioned in the second part of the reviewer remark, is performed in the first place to increase diversity in the test sets, instead of reducing its power. The distribution being heavily skewed towards a very few complexity labels, this filtering helps select problems where a diversity of solutions from a diversity of classes are present. In addition, the seed dataset, Deepmind Code Contest [1] contains many problems where corresponding i/o tests for checking correctness provide biased results, leading to high false positive rates or on the contrary high false negative rates. Filtering helps remove such problems that would hurt the evaluation of LLMs on these tasks. Finally, the test set creation is performed to preserve the diversity of problem difficulties and of problems subjects. On this test set, the task is challenging enough to differentiate model capabilities, and transparency on the filtering steps and failure analysis are provided to help in the creation of future, hopefully larger and better, datasets for these tasks.
>
> Concerning the complexity distribution, this is discussed in point 2 of reviewer 2 (in order to shorten this summary and centralize topics in the same threads).

---

> > ### Author Response · Authors · 2025-11-21
> >
> > > “The All@1 scores for complexity generation are exceptionally low (often <5% for time, <3% for space) across all models, including powerful reasoning models. While interpreted as LLM failure, these near-zero scores could equally indicate issues with the benchmark itself: unreliable ground truth, an overly harsh metric (All@k), or ill-posed tasks. It's hard to draw meaningful conclusions or measure progress when performance is near the floor.”
> >
> > Across the benchmark models and tasks, multiple metrics show that the progress is indeed measured: pass@1 on the complexity generation task goes from 10.6% for Codestral up to 33.5 for LLAMA 3.1 NEMOTRON-ULTRA 253B. All@1 more specifically, though obviously more challenging, scales from 1.3% up to 6.1% for the same models, which is a x6 performance gap. Looking at the appendix and the wider tables of results across more samples, All@10 reaches 26.1% with Qwen 3 32B (reasoning mode), proving that the task is learnable with more samples per prompt.
> >
> > Looking at the results per type of model (non-reasoning versus reasoning) or across time, we do observe the expected improvements. We underline that the added value of a benchmark lies in:
> > Its discriminative power, being capable of differentiating models through the assigned results
> > Its challenging power, AKA underlining tasks that are not mastered by LLMs currently and help shape the direction of improvements for future model releases.
> >
> >
> > > “Fine-tuning Llama 3.1 70B specifically on the benchmark tasks yielded negligible or even negative impacts on complexity generation performance (Table 4). This suggests the benchmark data and tasks, as currently formulated, may not provide a useful signal for improving LLM capabilities in this area, questioning their utility.”
> >
> > As explained in the response to the previous remark, the benchmark value is measured in its capacity to discriminate the performance of different models. In addition, looking at the improvements over time release of models, or across family of models (non reasoning versus reasoning, non code specific versus code specific), the benchmark does provide numbers that correlate the model improvements perceived.
> > Concerning the robustness of the benchmark to naive fine-tuning, this is in fact a quality that is underlined, as a quick labeling effort that would break it would mean poor value of adding the benchmark to the set of evaluations that already exist. The benchmark tests deep understanding and reasoning capabilities, not naive pattern matching that can be memorized during fine-tuning.
> > In addition, the benchmark does show on the more simple tasks, complexity prediction, some signal in being learnable: +8% improvement on time prediction and +18% improvement on space prediction. And a more complex task, like complexity generation, is tougher to learn.
> > As known limitations of fine-tuning, such trained models struggle at learning new reasoning patterns or compositional tasks: this is why we underline that pushing the RL that follows could be a source of improvements for harder tasks such as the ones introduced in this benchmark. Finally, as underlined in the discussion section, “new challenging benchmarks are perhaps more about finding out-of-distribution patterns of thinking rather than new data points of existing reasoning scenarios.”
> >
> >
> > > “The framework relies on dynamic analysis (profiling runs). It's unclear how well this empirical approach approximates theoretical worst-case complexity, especially given Python's dynamic nature and CPython optimizations. The benchmark might be evaluating the ability to generate code that performs well empirically under the framework's specific testing conditions, rather than code with a provably correct asymptotic complexity.”
> >
> > This is addressed in our response to comment 1 (for the sake of shortening our response, and centralizing similar topics in the same response thread).
> >
> > > “Decisions like using Big-O for worst-case, the specific parameters for fuzzing/curve fitting, and the simplicity bias need stronger justification and sensitivity analysis.”
> >
> > In terms of the justification of why Big-O was chosen for the worst-case, Appendix B does discuss in detail the definitions of the different notions of complexity in programming and why Big-O was chosen as the most relevant. In addition, Appendix C provides an ablation on different prompt formulations to explain the design choices made by the benchmark.
> > Appendix G does provide extensive details about the framework implementation. For further justifications and details about the framework design choices, we will release the entire codebase as part of the paper publication.
> > Generally speaking, we believe that the paper does cover a lot of details about our benchmark and we think that further detailing in the paper specific coding details would be out of scope and dilute the main results and contributions already reported.

---

> > > ### Author Response · Authors · 2025-11-21
> > >
> > > > “Could the authors provide a more detailed error analysis for the complexity framework? What types of code or complexity classes does it struggle with most? How was the 125-sample human validation set selected, and what was the inter-annotator agreement if multiple humans were involved? Given the 16-18% error rate, how confident can we be in the benchmark labels, especially for evaluating subtle differences between models or measuring fine-tuning progress?”
> > >
> > > We indeed conduct a human annotation effort to report potential error rates on the labels used in the test set. As reported in the remark of the reviewer, we measure a 16-18% error rate. In addition, main issues are due to unusual data types like tuples and triple nested lists that do have higher error rates. Finally, some edge cases not covered by the framework fuzzing range can account for some errors as well, and augmenting the framework with a wider set of tests, potentially conditioned on the problem description could be an area of improvement. We do highlight that the framework in its current state is completely LLM-free, so as to avoid any form of bias in the evaluation, in particular as the benchmark remains challenging even for the top performing LLMs.
> > > In addition, in an error analysis that looked at all coding problems available in the original dataset Code Contests [1], we found out that 84% of problems have a fail rate below 30% and only 4.5% of problems have a fail rate above 90%: for the first and majority types of problems, this is due for example to human solutions that do not compile or do not run, sometimes due to older python versions (the released human solutions can be of the times where Python 2 was used). For the second and minority part of problems, this can be due to exotic libraries needed for the problems to run, or improper design of tests that are used to check correctness (some problems have a set of tests that fail for any solution, be it correct or incorrect, that is run against them). And of course a portion of these failures remain the fault of the framework, for the above stated reasons. The heatmap breakdown in figure 3 provides further analysis of error rates per problem number of inputs and inputs types.
> > >
> > > > “How sensitive are the framework's complexity estimations to the specific hardware, Python version, background processes, and profiling tool versions? Were experiments run to quantify this variability? Could this noise explain some of the poor LLM generation results?”
> > >
> > > As detailed in the benchmark experimental setup, the experiments conducted in the paper were done on a cluster of h100 gpus and the python version was fixed to 3.10, and so must be the code being run so as to be compatible with the python version used by the sandbox environments. While running, CProfiler for the runtime measures and tracemalloc for the space measures were used, and ablations were performed on different profiling tools to measure which were the least sensitive to noise and variance across multiple runs. Prepared code and expanded inputs are run in independent Bubblewrap [2] sandboxes in order to prevent harmful effects. Resource management includes memory/time thresholds, CPU allocation policies according to the workers schedules, specific forkserver to control the scheduling of the sandboxes and handling of the sandboxes so to be fixed to specific cpus over the entire course of the runs.
> > >
> > > > “Can the framework distinguish between empirical performance fitting a curve (e.g., O(n) up to n = 10,000) and true theoretical complexity (e.g., an underlying O(nlogn) algorithm that looks linear in the tested range)? How does it handle amortized analysis?”
> > >
> > > As detailed in the Complexity Framework Parametrization appendix, measurement parameters such as range of values, step of value increases, multiple re-runs, etc are ablated in order to maximize framework performance. In particular, distinguishing between O(nlogn) and O(n) in certain cases of very slight linearithmic aspect was the most challenging, and various curve fitting methods including the use of a simplicity bias did help in this. In practice, accuracy limitations may arise when two classes exhibit very similar behavior over the typical range of fuzzing of the framework (1-10000).
> > > Throughout the study, the complexity refers to the worst case complexity, finding how input growths maximally affect runtimes and memory footprints. As the fitting of the complexity class remains depending on the curve of measures, if on average over a class of inputs of different sizes the runtimes keep a linear trend, this will be considered as a linear complexity class. So the framework does allow for amortization of punctual larger runtimes/memory footprints.

---

> ### Author Response · Authors · 2025-11-21
>
> > “Could the extensive filtering applied during test set creation (e.g., removing outliers, unstable predictions, unlikely ASTs) have biased the benchmark towards problems/solutions the framework can easily analyze, potentially masking harder cases?””
>
> The goal of the benchmark is to compare different models on a reasonably representative set of coding problems, in terms of their difficulty and topics. The test set reports these elements and we made sure to even stratify rare classes to make sure the test set is more representative and challenging (given how the distributions are typically skewed). In this setting, according to the reported model results, the benchmark retains its discriminative power and is far from saturated. We do not think there is any systematic bias that could alter the benchmark analysis of results.
>
> > “Given the near-zero All@k generation scores, how can the authors be sure this reflects inherent LLM limitations versus issues with the benchmark's noisy labels, task formulation, or the stringency of the All@k metric?”
>
> This is addressed in our response to point 3.
>
> > “Why do the authors believe fine-tuning failed to improve (and sometimes worsened) complexity generation performance? Does this suggest the benchmark data lacks a useful signal, or that standard fine-tuning is simply inadequate for this type of reasoning”
>
> This is addressed in our response to point 4.
>
> > “Why was Big-O chosen for the main benchmark despite the potential ambiguity noted, especially when the Big-Theta prompt seemed to improve generation for the reasoning model?”
>
> This is addressed in our response to point 6, in appendix B and appendix C. Appendix B.3 does explain in detail the difference between Big-O and Big-Theta, and why the latter would be ill-defined. In this context, BigO notation, used to describe the worst running case complexity of a piece of code (that is to say the tightest possible lower bound of this running case) turned out to be the most grounded notion of complexity to build a benchmark upon, and also the easiest to grasp for potential future users of the benchmark.
>
> **References**
>
> [1] Li, Yujia, et al. "Competition-level code generation with AlphaCode." Science 378.6624 (2022): 1092-1097.
>
> [2] Bubblewrap library. https://github.com/containers/bubblewrap, 2024.

---

### Official Review · Reviewer_Ka3Q · 2025-10-29

**Soundness:** 2
**Presentation:** 3
**Contribution:** 3
**Rating:** 6
**Confidence:** 4

**Summary:**

BIGO(BENCH) introduces a benchmark for testing whether LLMs can understand and produce code that meets stated time and space complexity. It covers three tasks: predicting the complexity of a given solution, generating a solution that satisfies a target complexity, and ranking solutions within the same class by constant factors. The labels come from profiling code across input scales rather than purely theoretical analysis.

Experiments show that today’s strong models often achieve functional correctness but still miss explicit complexity targets. Performance on complexity-constrained generation is low, and complexity prediction is only modest. The paper argues that current training does not teach models to control algorithmic complexity and calls for methods that align generation with resource goals.

**Strengths:**

1. The benchmark targets controlled time and space complexity rather than only functional correctness, and it instantiates this through three complementary tasks. The core idea feels novel, and the contribution is substantial and well scoped.

2. The authors annotate 3,105 problems and 1,190,250 Python solutions from CODE CONTESTS, provide per-input runtime and memory traces, and release problem-specific dataclasses so solutions can be profiled end to end. The test sets explicitly retain problems with multiple complexity classes, improving diagnostic value.

3. Single-sample performance on complexity-constrained generation remains low across models, and while larger sampling budgets help, the task stays difficult. Fine-tuning brings only targeted, partial gains, with All@1 often in the single digits, suggesting a genuinely hard benchmark that is likely to remain relevant over time.

4. The paper introduces a practical framework to infer time and space complexity from empirical profiling via fuzzing and curve fitting, and indicates an open release. It reports 84% agreement for time and 82% for space with human-theoretical labels, along with high self-consistency across runs.

**Weaknesses:**

1. Robustness of the inferred complexity labels is under-specified. While the authors report ≈90% self-consistency across multiple runs, it remains unclear how stable the labels are under runtime noise or varying sampling resolutions. For example, would the predicted class remain unchanged if input sizes were halved, or if 5–10% random noise were added to the timing measurements? A quantitative sensitivity analysis along these lines would strengthen confidence in the framework’s robustness.
2. The distribution is heavily concentrated in O(1) and O(n), which can inflate averages and conceal weaknesses on rarer classes like O(n log n) or O(n²). Class-balanced/re-weighted metrics, stratified subsamples, and per-class All@k would test whether gains persist beyond majority-class guessing and keep both tasks informative on the tails.

**Questions:**

1. You mention 37 algorithmic notions of  problems in your dataset. Could you provide the full list and their distributions?

2. In Table 2 several evaluated models are already outdated. Given newer releases such as o4-mini and GPT-5, how do you justify the current model selection? If immediate re-runs are not feasible, is there a plan for an updated leaderboard, submission interface, or fixed-budget evaluation protocol that would allow the community to add recent models while preserving comparability?

---

> ### Author Response · Authors · 2025-11-21
>
> We thank reviewer Ka3Q for the detailed feedback on our submission, and are pleased that the contribution of evaluating complexity beyond only correctness is appreciated.
>
> > “Robustness of the inferred complexity labels is under-specified. While the authors report ≈90% self-consistency across multiple runs, it remains unclear how stable the labels are under runtime noise or varying sampling resolutions. For example, would the predicted class remain unchanged if input sizes were halved, or if 5–10% random noise were added to the timing measurements? A quantitative sensitivity analysis along these lines would strengthen confidence in the framework’s robustness.”
>
> Many thanks for your remark about statistical noise in the runtime measurements. We take these remarks into account to update the Appendix G on the complexity framework, where we discuss the implementation details and the variance reduction techniques.
> When using the framework, the only entries are the code solution to evaluate and an example of inputs for the sake of initializing the framework with the types and forms of the inputs from the code snippet. Then, the scale of inputs being used for the fuzzing is determined by the framework itself, through various preliminary ablations that measured the impact of trying reduced or larger scales of inputs and their impact on the precision of the framework outputs. As reported in the paper, 91.9% (time) and 89.1% (space) of self-consistency are measured across 20 runs on 10,130 and 10,520 code solutions respectively, across many different cpus on a compute cluster, in order to account for statistical noise that may appear during these runs. During investigations of sources of noise, in order to make the framework better suited, we introduced for instance non-negative least squares curve fitting with a simplicity bias chosen to minimize the residuals. In addition, several variance reduction techniques, such as min-aggregation for runtime (3.5% improvement over median), using R=10 repetitions per measurement were also added. We agree on top of this that fuzzing for runtime and in general complexity inference is a whole area of research by itself, and we also suggest several improvements such as using Hodges-Lehmann estimator as potential enhancement.
>
>
> > “The distribution is heavily concentrated in O(1) and O(n), which can inflate averages and conceal weaknesses on rarer classes like O(n log n) or O(n²). Class-balanced/re-weighted metrics, stratified subsamples, and per-class All@k would test whether gains persist beyond majority-class guessing and keep both tasks informative on the tails.”
>
>
> As provided in the main paper and in the appendix, we agree that class-balanced/specific metrics do help better understand the performance, in particular for these tasks where the imbalance is an intrinsic nature of the underlying data.
> As stated in Section 4 about the data release, the data is naturally heavily tailed towards the most optimized complexity classes, as the data coming from competition platforms is indeed trying to solve the problems in the most optimized time and space settings. We report that linear time does represent 38% of all solutions, and constant time 20%. Whereas for space complexity the distribution is even more skewed with respectively 47% and 25%.
> In order to give a better understanding to future readers about the distribution of classes in the data, we also provide Figure 2 which shows the joint distribution of time and space labels. The test sets that are used for benchmarking were sampled so to upsample some rare classes, and therefore include 11 classes of time complexity and 5 classes of space complexity by being only ~300 problems large for each type of complexity.
> As part of the evaluation framework, metrics are macro-averaged per complexity class and then across problems, where each problem consists of at least 2 different classes of complexity. On top of macro-averaging the aggregated metrics reported in the main tables, we further provide more per-class analysis in Figure 4 and in the corresponding Discussion section: we confirm that models tend to under-perform on non-optimal complexity classes, compared to the most optimized class of every problem.

---

> > ### Author Response · Authors · 2025-11-21
> >
> > > “You mention 37 algorithmic notions of problems in your dataset. Could you provide the full list and their distributions?”
> >
> > Thank you for your suggestion, we are adding in the appendix a distribution of the algorithmic notions as present in the dataset. The 37 unique tags are: implementation, math, greedy, brute force, constructive algorithms, sortings, dp, strings, number theory, binary search, data structures, two pointers, combinatorics, graphs, dfs and similar, geometry, bitmasks, *special, trees, games, dsu, shortest paths, hashing, divide and conquer, probabilities, expression parsing, matrices, ternary search, string suffix structures, meet-in-the-middle, flows, graph matchings, fft, schedules, chinese remainder theorem, 2-sat, and finally the unspecified tag. The distribution is heavily skewed towards general algorithmic techniques, “implementation”, “math”, “greedy” and “brute force” being the most present subject tags. Across 3105 problems present in the training data, on average 2 tags are present for each problem, specifying for example a topic such as “math” or “geometry” and an algorithmic approach such as “data structures” or “two pointers”.
> >
> > > “In Table 2 several evaluated models are already outdated. Given newer releases such as o4-mini and GPT-5, how do you justify the current model selection? If immediate re-runs are not feasible, is there a plan for an updated leaderboard, submission interface, or fixed-budget evaluation protocol that would allow the community to add recent models while preserving comparability?”
> >
> > Several criterias were used to select the initial pull of models to be benchmarked: a mix of closed source and open source models (o1, gpt4, llama, codestral), from several institutions/companies, for which we can get access/funding to run the benchmark (within a reasonable budget), across a few iterations of model releases (llama 3/4, Qwen 2.5/3) and with and without reasoning (deepseek V3/R1). All of this by keeping the table of results within reasonable length to allow for a condensed presentation of the results and of the following discussion. In order to address newer releases, we have a code repository, data repository and leaderboard ready for release so that people can experiment with new models and contact us to display a new available score. In terms of comparability, we plan on allowing a fixed compute budget for the whole batch of test inferences (to be allocated per the model choices, which allow to highlight models that do smart compute allocation depending on the problem difficulty for example).

---

### Official Review · Reviewer_EQiS · 2025-11-03

**Soundness:** 2
**Presentation:** 3
**Contribution:** 3
**Rating:** 4
**Confidence:** 3

**Summary:**

The paper attempts to test how well code LMs understand the concept of code complexity by gauging how well they adhere to complexity-constrained code generation and code complexity prediction.

**Strengths:**

1. The paper uncovers an interesting setting in which Code LMs show a non-trivial lack of understanding.
2. The paper moves to improve the tooling and evaluation suite for open-ended code complexity evaluation by allowing arbitrary executable code to be profiled for runtime and memory usage over a large variety of inputs and then inferring its complexity via curve fitting.

**Weaknesses:**

1. The paper introduces a very specific task without really doing a good job of motivating its need. The need for generating the most efficient code is clear to all, but why does a model need to controllably generate code of a certain complexity is not very clear at all.
2. I would be sympathetic to a code understanding angle in terms of why controllable complexity is important, but I also feel there are better ways to disentangle that, for e.g. by asking the model to rewrite an existing code solution of a problem into another one that follows a specific complexity class.

Overall, my rating of 4 is harsh, and I would prefer to give a 5 if the option were present. I would also be open to be persuaded to move the score to a 6 if the work were better motivated.

**Questions:**

N/A

---

> ### Author Response · Authors · 2025-11-21
>
> We thank reviewer EQiS  for taking the time to review our paper. We are glad that the paper focus on a non-trivial lack of understanding of LLMs is found to be interesting.
>
> > “The paper introduces a very specific task without really doing a good job of motivating its need. The need for generating the most efficient code is clear to all, but why does a model need to controllably generate code of a certain complexity is not very clear at all.”
>
> We agree that better explaining the added value of our task can help improve the paper and are therefore sharing more context in an updated introduction.
>
> Beyond further evaluating the capability of LLMs at generating the most optimized code solutions, the goal of our task is first to identify whether LLMs are capable of distinguishing absolute runtime from complexity. During a preliminary analysis of Code Contests benchmark [1], we noticed that most i/o tests used to measure runtime remain of reasonable size, which means that in a setting with new larger i/o tests, solutions can rank very differently. In addition, during RL with runtime used as an objective, these same tests from Code Contests training set are used. Generating new tests is both an active and non-trivial field of research [2], and while such tests could be larger than the regular ones which could make them interesting, they can also slow down significantly the evaluation of code solutions during RL, which can make them impractical to train on. These reasons put together made us suspicious about the real capabilities of LLMs at optimizing code when outside of the code competition setting on which they were optimized during training and regularly compared to during benchmarking. In particular, we became interested in their true understanding of optimization at larger scale, which becomes more and more important as LLMs become heavily used for real-world engineering tasks. It is important not to forget that optimizing for small to reasonable size of inputs is not at all the same as optimizing for large inputs as in a production setting: a code can be faster than another on small inputs but way slower as inputs are above a certain large size. To bridge these limitations about i/o test based evaluation, we use complexity as a way to reliably know whether the optimization of a code solution is mastered. In addition, we leverage the coefficients of the complexity curve to explore the tradeoffs between different classes while looking at the in practice values of the curves, as going from linearithmic to linear complexity at the cost of way higher main coefficient may not be that interesting.

---

> ### Author Response · Authors · 2025-11-21
>
> > “I would be sympathetic to a code understanding angle in terms of why controllable complexity is important, but I also feel there are better ways to disentangle that, for e.g. by asking the model to rewrite an existing code solution of a problem into another one that follows a specific complexity class.”
>
> Many thanks for your remark. Following the first question about the intrinsic motivation behind why controllable complexity is important, we indeed gave some thoughts about the best ways to evaluate such understanding. To this end, the first task, complexity prediction, is designed to isolate the complexity understanding from the solution generation, by asking models to take an existing code solution and predict its complexity. We also report the pure code solution performance separately, to put aside the complexity requirements and verify whether LLMs can solve the problems in the first place. As underlined in Table 2, for example DeepSeek R1 achieves as high as ~70% pass@1 on pure code solution generation.
>
> Building on that, we chose to evaluate thirdly "Complexity Generation" as whether LLMs can both analyze code complexity and generate solutions at the same time, across different classes of complexity for a same problem. We believe that if the models have the capabilities in the first place to solve the problem, we can query them and report whether they can tweak the solution according to a complexity requirement, thereby measuring whether they truly understand code complexity.
>
> On top of this, we agree that tweaking an existing code solution into another could be an interesting task, though we are not convinced this brings much added value to the above described tasks, and to the arguments already made in the paper about the limited understanding of LLMs on complexity tasks. In order to make the benchmark more simple to grasp, we decided that two main complexity tasks, "complexity prediction" and "complexity generation", and a third more exotic task "complexity ranking", were already sufficient for getting the research community onboard on this realm of tasks. We would love to see other papers iterate on the complexity topic and try variations of tasks as suggested in your comment.
>
>
> **References**
>
> [1] Li, Yujia, et al. "Competition-level code generation with AlphaCode." Science 378.6624 (2022): 1092-1097.
>
> [2] Wang, Zihan, Siyao Liu, Yang Sun, Hongyan Li, and Kai Shen. "CodeContests+: High-Quality Test Case Generation for Competitive Programming." arXiv preprint arXiv:2506.05817 (2025).
>
> [3] DeepSeek-AI, et al. "DeepSeek-R1: Incentivizing Reasoning Capability in LLMs via Reinforcement Learning." arXiv preprint arXiv:2501.12948 (2025).

---

### Meta-Review · Area_Chair_jZEt · 2026-01-13

**Summary:**

This paper introduces BIGO(BENCH), a benchmark suite for evaluating whether code LMs can predict time/space complexity, generate solutions under explicit complexity constraints, and rank solutions by constant factors. Reviewers agree the topic is timely and valuable beyond functional correctness, and they appreciate the scale of the resource and the practical complexity-inference pipeline based on profiling, fuzzing, and curve fitting. One reviewer finds the benchmark concept substantial and likely to remain challenging, and another notes it exposes a meaningful gap: models can often solve problems but fail to reliably hit stated complexity targets. However, the overall rating towards reject because key aspects of benchmark validity and motivation remain insufficiently convincing. A major concern is that the benchmark’s ground truth relies on empirical complexity inference with non-trivial error rates against human-theoretical labels, sensitivity to profiling choices, hardware/runtime noise, and design parameters (fuzzing ranges, curve fitting, simplicity bias). Reviewers also highlight potential dataset bias from heavy framework-dependent filtering and class imbalance, and question interpretability when All@1 complexity-generation scores are near floor across models, making it difficult to separate model limitations from label noise or task stringency. The ACs recommend rejection.

**Reviewer Concerns:**

A major concern is that the benchmark’s ground truth relies on empirical complexity inference with non-trivial error rates against human-theoretical labels, sensitivity to profiling choices, hardware/runtime noise, and design parameters (fuzzing ranges, curve fitting, simplicity bias). Reviewers also highlight potential dataset bias from heavy framework-dependent filtering and class imbalance, and question interpretability when All@1 complexity-generation scores are near floor across models, making it difficult to separate model limitations from label noise or task stringency.

**Reviewer Scores:**

The rebuttal adds helpful implementation details, self-consistency statistics, and argues for discriminative power, but it does not fully resolve the central worry that the benchmark may be measuring fit the profiler rather than robust complexity understanding. The scores are 2, 4, 6.

---

### Decision · Program_Chairs · 2026-01-26

Reject